



# Mapping groundwater dependent ecosystems using a high-resolution global groundwater model

Nicole Gyakowah Otoo[1], Edwin H. Sutanudjaja[1], Michelle T. H. van Vliet[1], Aafke M. Schipper[2,3], Marc F. P. Bierkens[1,4]

[1]Department of Physical Geography, Utrecht University, The Netherlands
[2]Radboud University, Radboud Institute for Biological and Environmental Sciences (RIBES), Nijmegen, The Netherlands
[3]PBL Netherlands Environmental Assessment Agency, The Hague, The Netherlands
[4]Unit Subsurface & Groundwater Systems, Deltares, Utrecht, the Netherlands

*Correspondence to*: Nicole Gyakowah Otoo (n.g.otoo@uu.nl)

**Abstract.**

Global population growth, economic growth, and climate change have led to a decline in groundwater resources, which are essential for sustaining groundwater dependent ecosystems (GDEs). To understand their spatial and temporal dependency on groundwater, we developed a framework for mapping GDEs at a large scale, using results from a high-resolution global groundwater model. To evaluate the proposed framework, we focus on the Australian continent because of the abundance of groundwater depth observations and the presence of a GDE atlas. We first classify GDEs into three categories: aquatic (rivers

and lakes), wetlands (inland wetlands), and terrestrial (phreatophyte) GDEs. We then define a set of rules for identifying these different ecosystems, which are based, among others, on groundwater levels, and groundwater discharge. We run the groundwater model in both steady state and transient mode (period of 1979- 2019) and apply the set of rules to map the different types of GDEs using model outputs. For steady-state, GDEs are mapped based on presence or absence, and results are evaluated against the Australian GDE atlas using a hit rate, false alarm, and critical success index. Results show a hit rate above 80% for

each of the three GDE types. From transient runs, we analyse the changes in groundwater dependency between two time periods, 1979-1999 and 1999-2019 and observe a decline in the average number of months that GDEs depend on groundwater resources, pointing at an increasing threat to these ecosystems. The proposed framework and methodology provide a basis for analysing how global impacts of climate change and water use may affect GDEs extent and health.







## 1 Introduction

Global water consumption has quadrupled in the last century due to population growth and industrialization in areas with limited precipitation and surface water resources, increasing the dependency on groundwater resources (Kummu et al., 2016).

In addition, alterations in precipitation and recharge rates due to a changing climate have major impacts on groundwater resources (Cuthbert et al., 2019; Taylor et al., 2013). An increase in groundwater pumping and lower recharge rates have increased the rate of groundwater depletion in several regions globally (Bierkens & Wada, 2019). Overexploitation of groundwater resources by non-renewable groundwater use  in areas with low recharge rates leads to a decline in groundwater levels and a reduction of groundwater discharge to groundwater dependent ecosystems (GDEs) (Kløve et al., 2014).

GDEs are defined as ecosystems that are reliant on groundwater to maintain their ecological function and structure (Murray et al., 2006) (Kløve et al., 2014). The ecological integrity of GDEs depends on shallow groundwater levels or groundwater discharge, all year round, seasonally or periodically (Duran-Llacer et al., 2022; Foster et al., 2010). The degree of dependency of GDEs on groundwater varies with ecosystem type, geology, season, aquifer type, flow paths and catchment land use (Tomlinson & Boulton, 2010). In arid and semi-arid regions, groundwater is usually a major source of water for most

ecosystems. This dependency of ecosystems on groundwater includes surface water systems (aquatic GDEs, which include rivers and lakes) that rely on groundwater discharge (Kløve et al., 2011), and groundwater dependent wetlands and terrestrial ecosystems (e.g. vegetation like phreatophytes) that tap into groundwater as a source of water (Robinson, 1958).

It is evident that GDEs and their biodiversity and the ecosystem services they provide are at risk due to unsustainable groundwater extractions (Bierkens & Wada, 2019; Link et al., 2023). It is, therefore, necessary to implement protection

measures through groundwater management policies, such as the extension of buffer zones around groundwater recharge zones and appropriate land management in groundwater capture hotspots (Kløve, Balderacchi, et al., 2014; MacKay, 2006). A critical step towards the large-scale application of these water management strategies is to better understand their global distribution and response to environmental change. This, in turn, requires delineating the global spatial distribution and extent of GDEs, understanding temporal variations of the dependency of these ecosystems on groundwater and assessing how they are impacted

by sectoral groundwater withdrawals.

Until the past decade, mapping of GDEs was predominantly done at local scales, through laborious and costly methods that involved long hours of field surveys (Eamus et al., 2006; Hatton & Evans, 1998). More recently, GDEs have also been mapped based on satellite imagery such as MODIS (Castellazzi et al., 2019). Some large-scale satellite imagery-based mapping studies (> 50km) have been done in Chile (Duran-Llacer et al., 2022), Colorado and Nevada (Werstak et al., 2012), California (Howard

& Merrifield, 2010), The Netherlands (Bonte et al., 2013; Hoogland et al., 2010), Ireland (Kilroy et al., 2009), South Africa (Münch & Conrad, 2007), Spain (Martínez-Santos et al., 2021; Münch & Conrad, 2007) and Australia (Barron et al., 2014; Brim Box et al., 2022; Glanville et al., 2016). The first continental mapping was done for Australia (Doody et al., 2017), combining remote sensing, GIS and expert knowledge to create a GDE atlas for the continent.





All the studies mentioned above are static in the sense that they map the spatial distributions of GDEs at a given point (or year)
in time. However, to understand the dynamics of these ecosystems, it is essential to develop a method that can simulate changes
over time. The use of machine learning to predict groundwater dependency by ecosystems is a promising tool for spatial
simulations. However, little data and an insufficient understanding of catchment-scale dynamics limit the use of machine
learning (Xu & Liang, 2021) for mapping spatio-temporal GDE dynamics. Process-based groundwater flow models, preferably
at high resolution, may be more suitable for spatio-temporal mapping of GDEs, since they enable explicit linkages between
GDE expression and groundwater level and groundwater discharges. In addition, process-based groundwater flow models have
the potential of scenario analyses, i.e. to be applied under various assumptions of future changes in climate, land use and
human water use, which all may impact future changes in GDE extent (Fatichi et al., 2016). This was first shown by De Graaf
et al. (2019) using a global groundwater model to project changes in groundwater discharge to streamflow. It is also possible
to couple a process-based dynamic GDE mapping model to other model types such as a biodiversity or economic models to
determine the relationship between GDEs and biodiversity or the values of ecosystem services (Barbarossa et al., 2021; Van
Emmerik et al., 2014).

The aim of this research is to explore the potential of mapping the spatio-temporal dynamics of GDEs based on a global
groundwater model. This work builds on the earlier work of De Graaf et al. (2019) in that it considers a wider range of GDEs
and uses a much higher resolution groundwater model. We first classify GDEs into aquatic, wetlands and terrestrial vegetation
(phreatophytes) ecosystems (section 2.1). We then use a global coupled surface - groundwater model run at 1km resolution in
steady state and transient mode (section 2.2) to map the distribution of these three GDE classes in Australia (section 2.3). We
also analyze the temporal variations for the three different GDE types (section 3). We choose to focus on Australia because of
the availability of an existing GDE atlas (Doody et al., 2017) and the abundance of extensive groundwater monitoring data,
which enable us to evaluate our method and results. Also, Australia has a large variation in hydro-climatology and topography,
which will enable us to understand the potential of our developed framework and methodology in various landscape settings.

## 2 Data and methodology

This section is divided into subsections highlighting the entire GDE modelling framework which entails model set-up and
evaluation, GDE classification and temporal variation analysis. The framework for mapping GDEs is presented in Fig.1. Using
this framework, firstly we define the GDE classes (step 1), then we run the surface-groundwater model and evaluate the
groundwater levels against well observations (step 2). Finally, we use the model output to map and analyze and evaluate the
spatio-temporal mapping of the three different classes of GDEs (Step 3).





**Figure 1: Groundwater dependent ecosystems (GDE) modelling framework using a high-resolution groundwater model.**





## 2.1 Defining GDE classes (step 1)


We categorize groundwater dependent ecosystems into three classes based on interaction with groundwater (see Figure 2). These include 1) ecosystems that depend on sufficient groundwater discharge (aquatic GDEs such as streams, rivers and lakes); and 2) ecosystems that need shallow groundwater tables and soil saturation (wetland GDEs); 3) ecosystems that depend on groundwater for root water uptake (terrestrial GDEs with phreatophyte vegetation). Note that we focus on only inland

ecosystems only.

For aquatic GDEs, any stream or lake pixel where the ratio of groundwater discharge ($Q_{gw}$) to total streamflow ($Q$) $\frac{Q_{gw}}{Q} > 0$ for more than a month is classified as being groundwater dependent. The rationale behind using groundwater discharge as a metric is that it maintains streamflow during dry spells and due to the relatively constant temperature of groundwater, which modulates stream temperatures during warm periods.

For terrestrial GDEs (phreatophyte vegetation) we assume that any cell with a vegetation type with maximum rooting depth ($D_{rmax}$) greater than the groundwater level of that cell is groundwater dependent, assuming that in this case, the vegetation is able to access groundwater with its deepest roots during dry spells.

We define wetland GDEs based on the fraction of saturated area (soil wetness) and groundwater level. Any cell that has a saturated area fraction ($F_{sat}$) greater than 50% and a shallow groundwater table (less than 5m) is classified as a wetland GDE.

The 5 meter groundwater level threshold was obtained from (Gerla, 1992), while the 50% soil saturation was added to discern dry areas with shallow groundwater levels from actual wetlands that are typically situated in areas with topographic convergence. Note that we can make this classification based on steady-state groundwater model results as well as based on transient model results. In the latter case, we assess the "*degree* of groundwater dependency" for each GDE type identified at a monthly time step.

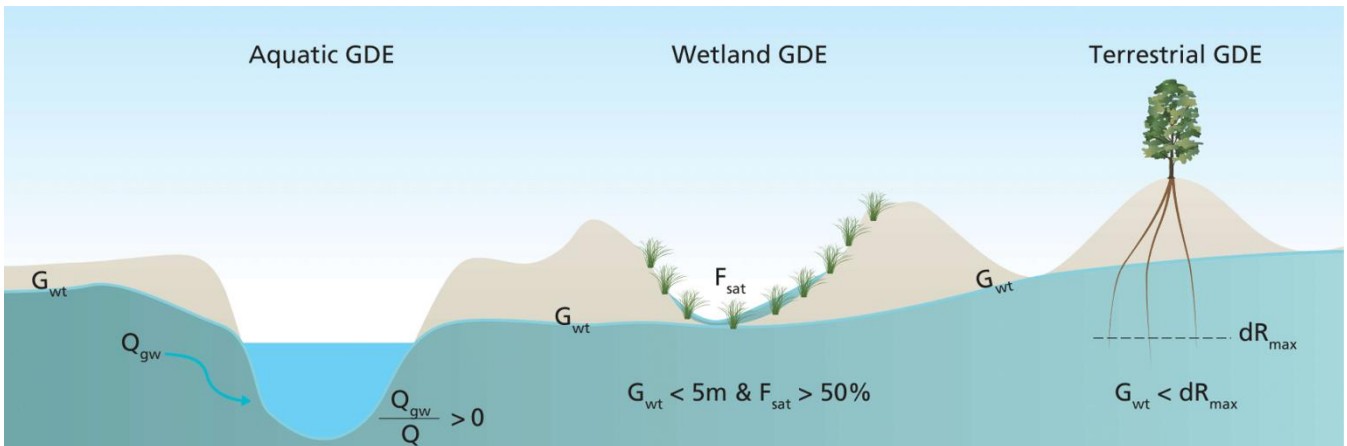


**Figure 2: Criteria for defining ecosystem dependency on groundwater with $Q_{gw}$ local groundwater discharge, Q accumulated stream flow, $G_{wt}$ = groundwater depth, $F_{sat}$= saturated area fraction and $dR_{max}$ = maximum rooting depth.**




## 2.2
## Model set-up, sensitivity analysis and output evaluation (step 2)

For this research, we use an integrated hydrological model that consists of two parts. The first part is a global physically-based global hydrology and water resources model (PCR-GLOBWB version 2.0) (Sutanudjaja et al., 2018) that simulates global

terrestrial hydrology including the human impacts (dams and human water use). The second is a time-dependent (transient) groundwater flow model (Verkaik et al., 2022). The two models are linked through a one-way coupling, that is, the outputs of the PCR-GLOBWB model are used as inputs to the groundwater flow model (Sutanudjaja et al., 2011). We first run the PCR-GLOBWB 2 with the default groundwater model and then use the time series outputs for surface water levels, saturated area fraction, and groundwater recharge as forcing for the groundwater flow model.

**2.2.1 PCR-GLOBWB**

PCR-GLOBWB 2 is a gridded integrated hydrology and water resources model with a latitude-longitude grid of 5 arcminutes spatial resolution that simulates terrestrial hydrology and human water use at a daily time step. A detailed model description can be found in Sutanudjaja et al. (2018). PCR-GLOBWB 2 is forced with precipitation, temperature and reference evaporation based on the W5DE5 (Cucchi et al., 2020; Lange et al., 2021) meteorological data set. Soil parameters are based on the

SoilGrids (Hengl et al., 2017). The default model settings of four landcover types were used, aggregating land cover classes into tall natural vegetation, short natural vegetation, non-paddy irrigated crops and paddy irrigated crops (Sutanudjaja et al., 2018). To simulate variations in the saturated area fraction, the Arno Scheme (Todini, 1996), which is an integral part of PCR-GLOBWB 2 to assess the area subject to surface runoff, is utilized. PCR-GLOBWB 2 also has an irrigation and water use model that calculates water demand (Wada et al., 2014) and water withdrawal, water consumption and return flows for

irrigation and domestic, livestock and industrial sectors.

**2.2.2 Groundwater model**

We use a two-layer groundwater model run at 30 arcsec (16 million active cells) to simulate groundwater depths, groundwater heads and groundwater discharge. The model code that is used is MODFLOW 2005 and the aquifer properties are taken directly from (de Graaf et al., 2017). The groundwater model is forced with surface water levels and net groundwater recharge

(percolation minus capillary rise) over the period 1979-2019 at monthly time steps as obtained from runs with PCR-GLOBWB 2. For net recharge simple resampling is used, while water levels are computed at 30 arcsecs based on a simple routing (method of characteristics, for details see Sutanudjaja et al., 2018) of the 5-arcminute specific discharge over a 30-arcsecond drainage network based on Hydrosheds (Lehner et al., 2008). The steady-state groundwater model is run with average net groundwater





recharge and surface water levels over 1979-2019. Consequently, the transient run follows with the heads from the steady state
run as the initial condition.

### 2.2.3 Sensitivity analysis and calibration of groundwater model parameters

With groundwater recharge and boundary conditions as described above, the groundwater model results are possibly sensitive
to aquifer transmissivity and storage coefficient, river bed conductance and the thickness of the confining layer, while these
properties are often very uncertain at larger scales (Brunner et al., 2017). We perform a sensitivity analysis using 216 steady-
state simulations varying the following three parameters: riverbed conductance, vertical conductivity of the confining layer (if
present) and transmissivity of the confined and unconfined aquifers. We change these parameters independently using a single
prefactor $k$ applied to the log-transformed parameter of concern, with $k$=1 the initial value of the parameter taken from De
Graaf et al (2017). See Eq. (1) for an example for the transmissivity:


$$T' = \exp\left(k \cdot \ln(T)\right) \tag{1}$$

With $T'$ the perturbed transmissivity ($M^2$ $d^{-1}$), $T$ the original transmissivity according to De Graaf et al. (2017) and $k$ the
prefactor applied.

For each unique parameter combination, we evaluate the biases between the simulated steady state groundwater depth (surface
elevation minus hydraulic head in the top layer) and time-averaged observed groundwater depths using data from 15,345 wells
recorded from 1970 to 2019 at monthly time step. If there were multiple wells within a 1km cell, we take the average of these.
We then select the best parameter set with the least bias against observed well data and vary the storage coefficient and conduct
six transient runs to select the best parameter set for simulating transient groundwater levels. Based on this, we finally select
the best parameter set for the GDE mapping.

### 2.2.4 Evaluation of simulated groundwater depths

We evaluate the transient simulated groundwater depths against observed groundwater well depth time series data (BOM
AU,2023). A total of 5 million cells with simulated groundwater depths are evaluated against the observed data from 1979 to
2019 in the Australian continent. The metrics used for evaluation are bias (Baker, 1987), Pearson correlation coefficient (Cohen
et al., 2009) and  relative variance (Grömping, 2007).





## 2.3 GDE mapping (step 3)

### 2.3.1 Steady-state GDE mapping

After running the model in steady state (average forcing groundwater dependent), we map the three different classes of GDEs according to the classification rules described above (Figure 2). For aquatic GDEs, we derive an aquatic ecosystem dependency ratio to groundwater which is defined as $\frac{Q_{gw}}{Q}$ where $Q_{gw}$ is the local groundwater discharge and $Q$ is the total streamflow.

Wetland GDEs are mapped using the groundwater depth from the groundwater model and the average saturated area (1979-2019) from PCR-GLOBWB 2. Terrestrial vegetation GDEs are mapped using the groundwater depth and a rooting depth map

(Fan et al., 2017).

After mapping these GDEs in steady state we evaluated the results by comparing these with the GDEs mapped by the Australian GDE Atlas using similarity index metrics. These metrics are the hit rate *h* (a class is present that is also mapped), false alarm rate *f* (a class is mapped that is not present) and miss rate *m* (a class is present that is not mapped). From these metrics, we also calculate the critical success index (CSI) for each mapping of each GDE type defined as Eq. (2):

$$\text{CSI} = \frac{h}{h + f + m} \tag{2}$$

Note that the Australian GDE Atlas distinguished between actually observed GDEs and likely GDEs (Doody et al., 2017), where the latter are mapped based on landcover type. When evaluating the mapping, we did not distinguish between known and likely GDEs..

### 2.3.2 Transient GDE mapping

For the mapping of transient GDEs, we use monthly time series of groundwater depth, groundwater discharge and saturated and saturated area fraction from the transient simulation over the period 1979 – 2019. We use the same criteria for mapping GDEs as used for mapping in a steady state and use the extent of the steady state mapped GDEs as a given. Within these areas we then consider the temporal variability in the contribution of groundwater as well. For aquatic GDEs we use monthly values of $\frac{Q_{gw}}{Q}$ to classify each month as low-dependent (ratio < 0.25) moderately dependent (ratio between 0.25 and 0.75) and highly

dependent (ratio > 0.75) on groundwater. For terrestrial and wetland GDEs, we record the average number of months per year that the system is classified as groundwater dependent. We separately identify these transient GDE measures for two 20-year periods (1-1-1979 to 31-12-1999 and 1-1-2000 to 31-12-2019) to assess potential changes in the contribution of groundwater between these two time periods.





## 3. Results

We first present the evaluation of the groundwater model simulations used as a first performance indicator of the proposed GDE mapping methodology (section 3.1). We then evaluate the coincidence of GDE types mapped with the steady-state groundwater model with GDEs mapped by the Australian GDE atlas (Doody et al., 2017) (section 3.2). Finally, we show the temporal change in the degree of groundwater dependency of the different ecosystem classes based on the transient simulations over the period groundwater dependent (section 3.3). Further information on the simulated groundwater levels, evaluation
metrics and sensitivity analyses can additionally be found in the S3.

### 3.1 Performance of the groundwater model in simulating groundwater depth

From the sensitivity analysis and calibration, it turned out that in this case the performance metrics calculated from the groundwater level observations where rather insensitive to the pre-factors (see Figure S3). We therefore decided to use the default parameters for further analyses. In general, the cumulative frequency distributions show a good agreement in timing
(~ 75 % shows r > 0.25). The dissimilarities between the observed and the simulated head is due to the bias. Our simulated heads are deeper than the observed, however with ~ 70 % having a bias ranging from 0 to 5 m. Plotting a scatter of the bias against the simulated heads (Figure S4), we observe a smaller bias for shallower depths compared to the deeper depths. This shows that where it matters for GDEs, the biases are also smaller. The relative variance shows an underestimation of groundwater level variation ~ 80% with a relative variance < 0. 6).

The groundwater depth of the first layer as simulated with the steady-state groundwater model and the best parameter set from the sensitivity analyses is shown in Supplementary Information Figure S1 presenting a high range in groundwater depth over Australia (0.25m to >320m). Figure S2 shows the differences in simulated (steady state) groundwater heads for areas where a confining layer is present. The red areas are those where there is a confining layer and the heads in the aquifer underlying the confining layer are larger than those in the confining layer itself. In these areas, it is possible that deep incising surface waters
could receive groundwater discharge from the lower aquifer..

Figure 3 shows maps as well as cumulative frequency distributions of the bias (in m; difference in temporal mean depths: simulated minus observed), Pearson correlation coefficient (between the observed and simulated time series over time) and the relative variance (temporal variance of simulated time series divided by the temporal variance of the observed time series). Note that we compared the simulated groundwater depths from layer 1 with all available observation wells. Due to a lack of
data on the wells' filter depths, we were not able to exclude the wells with filters in confined aquifers. This will likely have a negative effect on model performance. Results show that the evaluation metrics perform better in Tasmania and areas where the wells are likely not in a confined aquifer, i.e., the red areas in Figure S2 (with r > 0.6, bias <= 3 m)).



## 3.2 Steady state mapping and evaluation of GDEs

To map the locations of GDEs we use the steady-state outputs from our groundwater model. For the aquatic GDEs, we observe

that most streams of well-known river basins such as the Darling River depend on groundwater. Quite some vegetation located in dry areas tap into groundwater levels, while wetlands, showing large ranges in size, depend on groundwater predominantly when located close to rivers, likely being wetlands in or nearby floodplains.

Evaluating our mapped GDEs against the Australian GDE atlas Doody et al. (2017) we observed a high hit rate and critical success index (CSI) of 87%, 92% and 95% for aquatic, terrestrial and wetland GDEs, respectively (Figure 4). Despite the

overall bias observed in the groundwater model (Figure 3), the impact on representing GDEs is limited since this bias is smaller for shallower groundwater levels than for deeper groundwater levels (Figure S4). For the aquatic GDEs, most of the false alarms are in the near coastal areas and also in the Great Artesian basin. We miss some groundwater dependent terrestrial vegetations (phreatophytes) in western Australia due to a lack of good rooting depth data. We also wrongly identify a large area of wetlands in New South Wales.

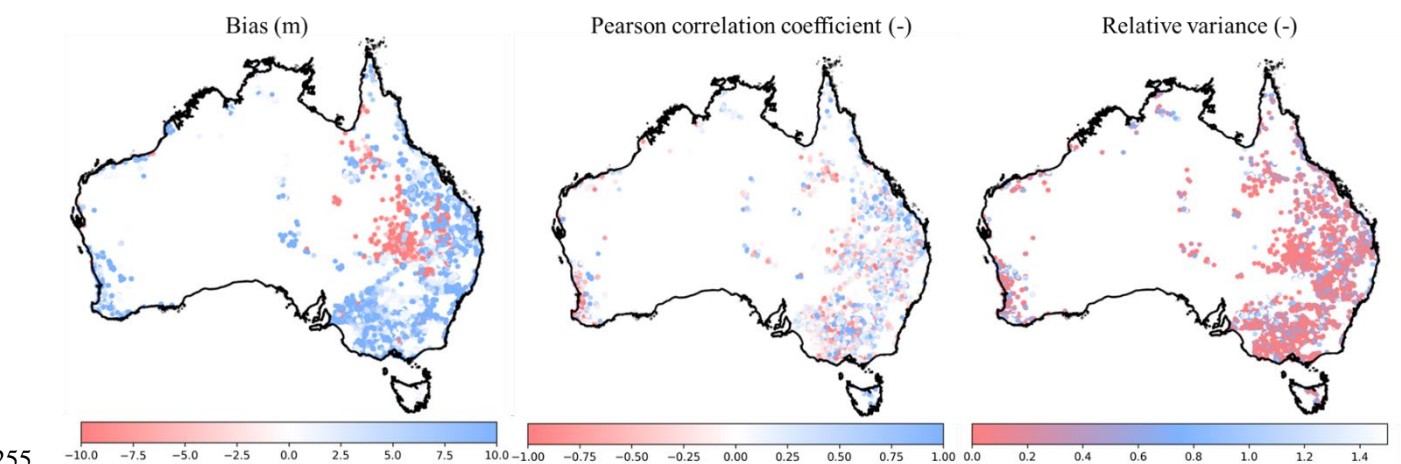


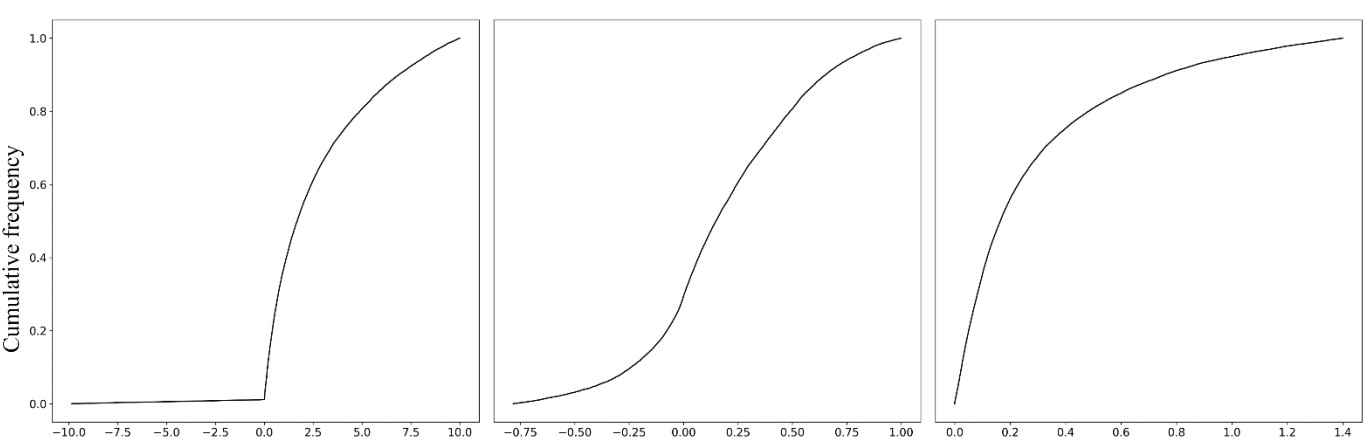





**Figure 3: Evaluation statistics of observed groundwater depths against simulated groundwater depth; top row: maps with values per observed location; bottom row: associated cumulative frequency distributions; left column: Bias (m); middle column: Pearson correlation coefficient; right column: relative variance. The white areas on the maps are locations without observation data.**

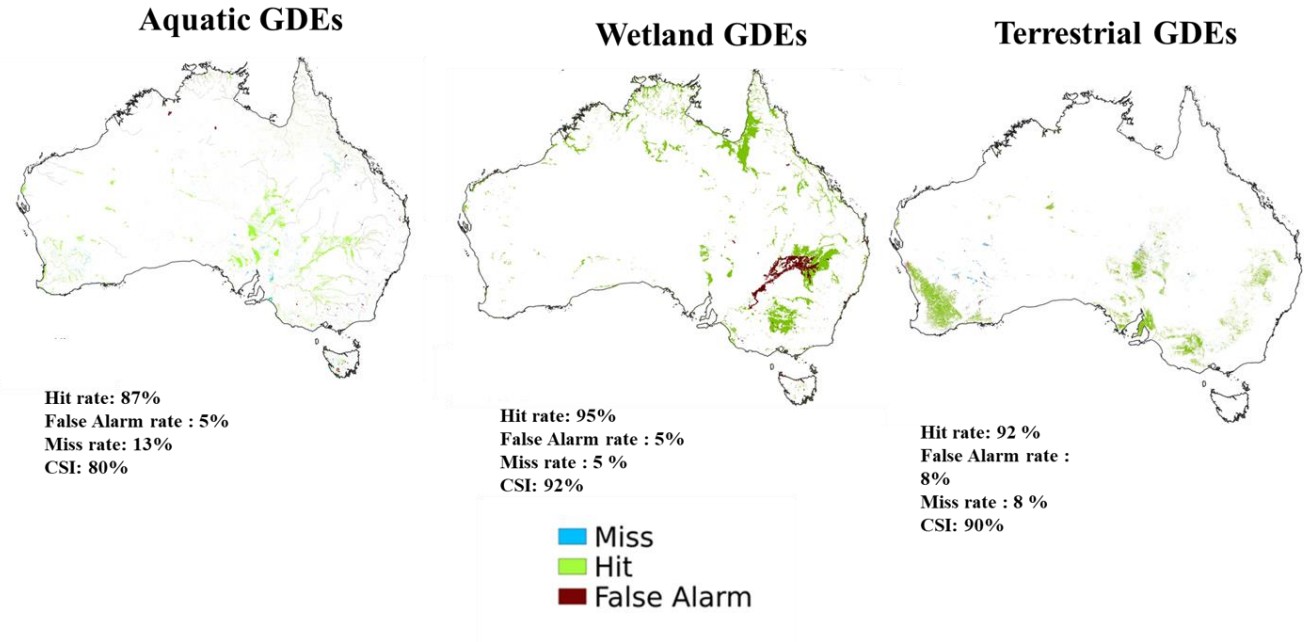


**Figure 4: Mapped GDEs based on steady state groundwater model results evaluated against the Australian GDE atlas showing hit rate, false alarm ratio, miss rate and the CSI for the three GDE classes. Blue colour represents missed ecosystems, red represents false alarm and green hit rate.**

### 3.3 Transient GDE mapping

To understand how the dependency of different ecosystems varied in the past, we divided the simulated periods into two-time intervals (period 1: 1979-2000; period 2: 2001-2019) and estimated for each time interval the average number of months per year that each GDE type relies on groundwater. Next, we calculated the changes in number of months of groundwater dependency: period 2 minus period 1 (Figures 5 and 6). As stated in the Methods, we used the mapped steady state extent as a given for the evaluation of the degree of groundwater influence on GDEs for the transient runs. In other word, we did not look

into extent dynamics.

For aquatic GDEs, we assessed temporal dependency in the different dependency ratio $\frac{Q_{gw}}{Q}$ categories. We observe that there is a decline in the average number of months in all dependency classes ecosystems (Figure 5) and that the decline in groundwater contribution is mostly observed in streams in the Murray Darlin Basin. This is in accordance with the decline in groundwater levels between the two periods in both the simulations and the observations (Figure S5). It is important to realize

that the dependency ratio depends on both the groundwater depth and related groundwater discharge $Q_{gw}$ and the streamflow





itself. This is illustrated in Figure 6 that shows simulated time series of $\frac{Q_{gw}}{Q}$, groundwater depth and total streamflow. The figure shows that the groundwater levels are constrained at the top by the drainage system and also shows the intermittent character of the Australian climate, with wet periods alternating with dry periods where groundwater levels decline, and streamflow becomes almost zero. The top figure shows a negative trend in groundwater levels. However, since streamflow is

also declining, these offsets the decline in groundwater discharge, resulting in a smaller negative trend in groundwater dependency $\frac{Q_{gw}}{Q}$. The zoom at the bottom shows the importance of discharge variability. November 2005 and July 2006 show almost the same shallow water table. However, streamflow peaks in November 2005, which makes for a low dependency ratio, while the 2006 streamflow is low in July, making the dependency on groundwater discharge large.

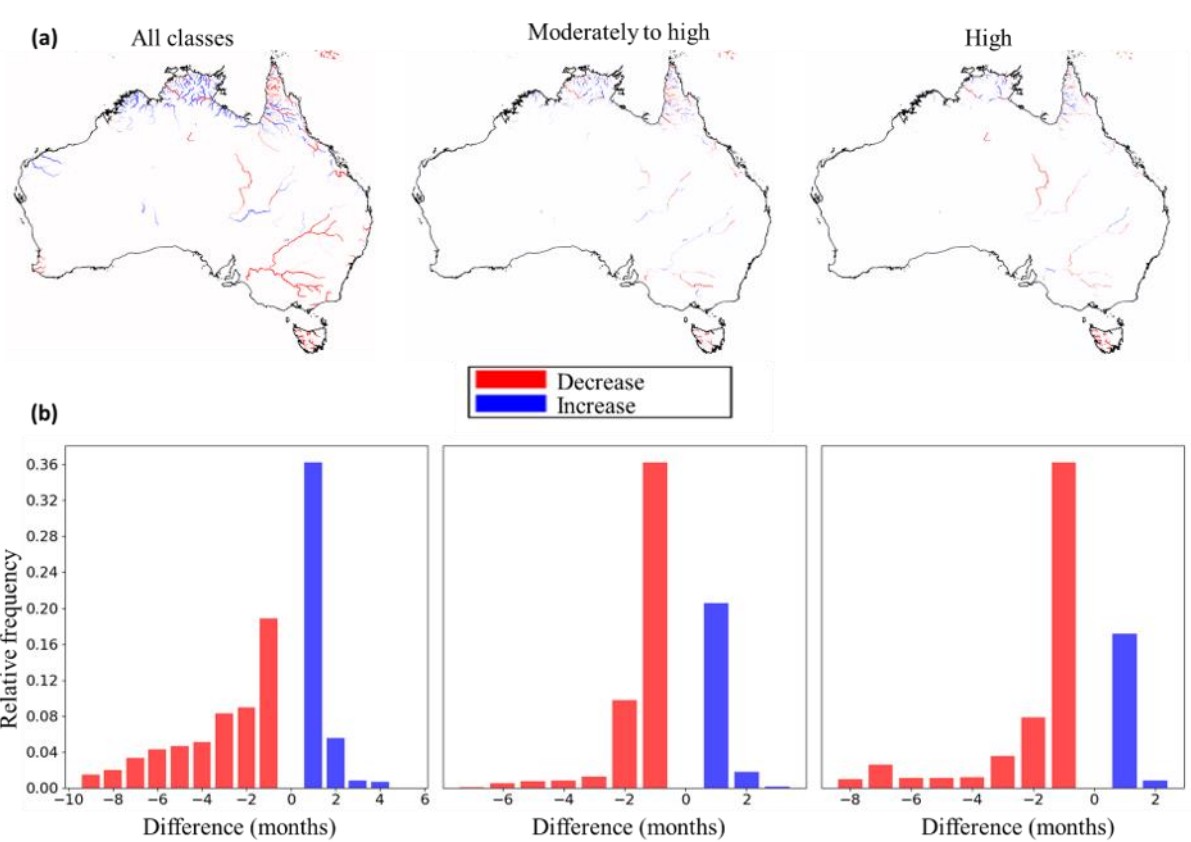

**Figure 5: Change in groundwater dependency of aquatic GDEs between 1979-2000 and 2001-2019; (a) Maps of the direction of change in the average number of months that aquatic GDEs depend on groundwater; the left figure shows the change in the number of months $\frac{Q_{gw}}{Q} > 0$ (low to high dependency), the middle figure $\frac{Q_{gw}}{Q} > 0.5$ (moderate to high dependency) and the right figure $\frac{Q_{gw}}{Q} > 0.75$ (high dependency); Red areas indicate a decrease in the average number of months with groundwater dependency and blue indicates an increase; (b) associated frequency distributions of change in number of months.**




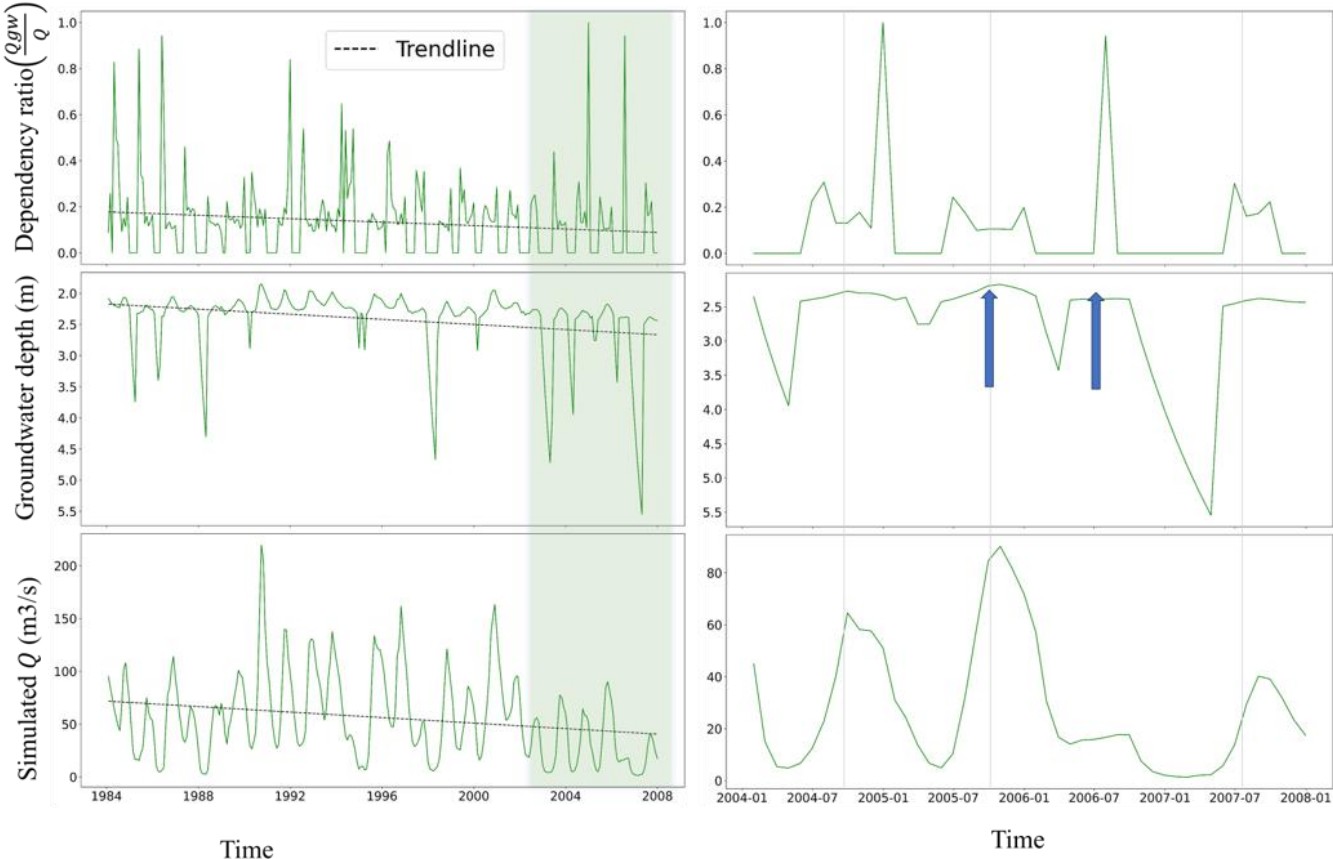

**Figure 6: Example time series of $\frac{Q_{gw}}{Q}$ for a downstream location in the Darling River. Top: time series of simulated $\frac{Q_{gw}}{Q}$, total streamflow ($Q$) and groundwater depth, including trendlines. Right: zoom into a selected timeframe (green bar in the left figure) to show how the variability of dependence of $\frac{Q_{gw}}{Q}$ depends on groundwater level streamflow.**

Figure 7 shows the change in the number of months that the terrestrial GDEs and wetland GDEs are groundwater dependent. For wetland GDEs, we observe a decline in groundwater contribution of on average four months per year in most regions, with an exception in some wetland areas in New South Wales and South Australia where an average increase of eight months of groundwater dependency is observed. For wetland GDEs, this decline can also be caused by a decline of the saturated area fraction, which is a driving factor for the decrease in wetland GDE dependency in Central Australia since these areas show

only limited declines in groundwater levels. Terrestrial GDEs (phreatophytes) show a limited decline in groundwater dependency of one month on average for most locations. These are small changes and can only be attributed to a decline in groundwater levels since the rooting depth is kept constant (See Figure S5).



We have added some additional analyses to improve understanding of the drivers' groundwater level changes between both
periods. Figure S6 shows the difference in simulated groundwater recharge between the periods 2001-2019 relative to 1979-
2000 and the simulated groundwater withdrawal over the 2001-2019 period. The changes in groundwater recharge reflect the
impact of climate variability and/or change on the groundwater system while the locations with groundwater withdrawal reflect
the direct human impacts. A thorough factor analysis is beyond the scope of this study, but a comparison of Figures S5 with
S6 suggests that climate variability mainly explains the changes in groundwater depth in Central and Western Australia while
both factors play a role in Eastern Australia. Note that the variability of the simulated groundwater levels is half of that of the
observed ones. This reflects the underestimation of the variability in groundwater depth as shows in Figure 3. Possible
explanations for this are an underestimation of recharge and recharge variability in drylands (Quichimbo et al., 2021), an
overestimation of storage coefficients and an underestimation of groundwater withdrawals.

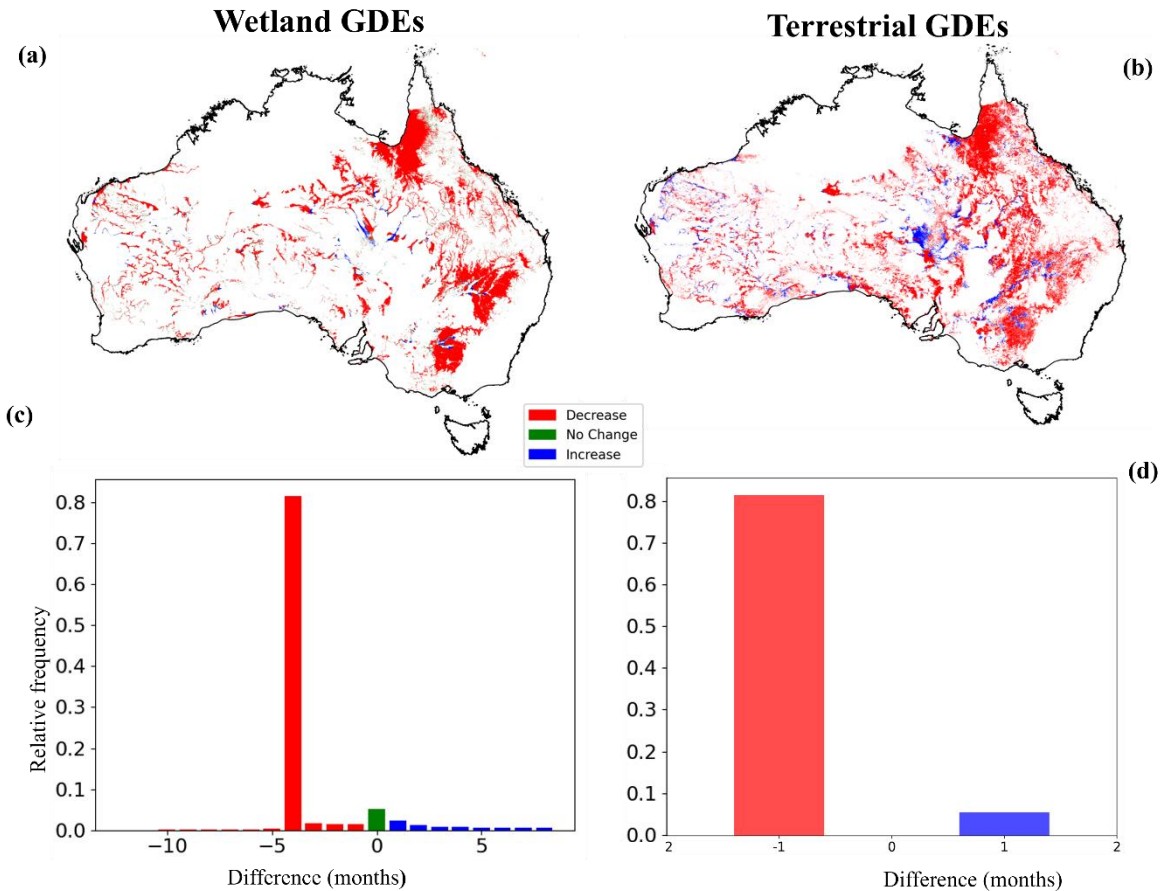


**Figure 7: Change in average number of months of groundwater the dependency of terrestrial GDEs (phreatophytes) and wetland
GDEs; a) direction of change in terrestrial GDEs (phreatophytes); b) direction of change wetland GDEs. Red areas indicate a
decrease in the average number of months with groundwater dependency, green indicates no change between the periods and blue
indicates an increase; (c) and (d) associated cumulative frequency distributions of change in number of months.**




## 4.0 Discussions and conclusions

In this research, we developed and evaluated a framework using a high-resolution surface-groundwater model at 30 arcsec resolution to map aquatic, wetland and terrestrial groundwater-dependent ecosystems. We evaluated the simulated groundwater depth with observed groundwater level observations and the mapped GDE occurrence with the GDE atlas of
Australia. Groundwater resources are crucial for GDEs as they partially or fully contribute to their water budget. Analysing the spatial and temporal changes in groundwater dependency is required for understanding threats to GDEs. In the context of global population growth, industrialization, economic growth and climate change driving global groundwater depletion, this will inform relevant stakeholders on high-risk ecosystems and direct groundwater allocation. This study introduces a method for GDE mapping that offers the possibility to improve understanding of the spatial distribution and temporal dynamics of
GDEs in relation to the spatiotemporal dynamics of groundwater systems.

Our research builds upon previous work on mapping GDEs combining expert knowledge, GIS and field visit by (Doody et al., 2017), previous global groundwater modelling efforts (De Graaf et al. 2019), as well as work by Eamus et al., (2015), who investigated GDEs responses to changes in groundwater depth using satellite images and field studies for selected locations. In comparison, our research proposes a methodology to understand the long-term temporal responses of different GDE types
to changes in groundwater levels at a large spatial extent and at fine resolution. Our method relies on outputs from a process-based high-resolution large-scale groundwater model and has potential for identifying hotspots of ecosystems threatened by groundwater extractions on a large scale. It proved to be effective for identifying GDEs in Australia with a hit rate over 80%. GDEs occur in areas with a shallow water table and, notably, our framework was able to simulate groundwater depths at these locations well. The transient component of this methodology also facilitates in-depth understanding of the temporal dynamics
of the reliance on groundwater resources by GDEs. At a monthly time, scale, we were able to simulate the different levels of dependency by aquatic GDEs as well as the levels of reliance or non-reliance on groundwater resources by wetlands and phreatophyte communities.

It is important to note that the dependency ratio of aquatic GDEs is dependent on both total streamflow and groundwater depth. Thus, increased groundwater discharge coupled with a decrease in streamflow may shift a river section to be more dependent
and vice versa. Although streamflow and groundwater levels are likely positively correlated at larger time scales, they may not be in phase at shorter time scales due to the different response times of surface water and groundwater systems. This makes the degree of groundwater dependency of Aquatic GDEs more intermittent than other GDEs that rely on groundwater depth and soil wetness (Wetlands) or groundwater depth only (and.phreatophyte communities) . Phreatophytes may be even more resilient to change as they are able to adapt to groundwater level declines through deeper rooting (Naumburg et al., 2005),
although there are limitations to this adaptive capacity between species, implying that a decline in groundwater level may result in changes in phreatophyte community composition (Sommer & Froend, 2014).



The model performance evaluation in the transient analysis revealed a fair overall agreement with observed groundwater depth data, yet also an overall overestimation in simulated groundwater depth. However, since biases for shallow groundwater levels were limited, the performance in identifying the GDEs was very good, as indicated by the different performance metrics. The calibration results show that the groundwater model was not very sensitive to global changes in parameter sets (Figure S4). This calls for more sophisticated groundwater calibration methods that allow for regional differentiation in model parameters. Also, further improvements can be expected if the recharge simulated with PCR-GLOBWB 2 could be better constrained. Therefore, a calibration approach more sophisticated than pre-factor parameter change must be implemented to improve the groundwater model simulations and derived mapping of GDEs.

One of the limitations of the current groundwater model setup is its it relatively simple hydrogeologic schematisation obtained from De Graaf et al. (2017). Although this makes the framework globally applicable, it may suffer from a lack of geological detail needed for representing groundwater discharge and springs over e.g., the Great Artesian Basin. Another limitation is the assumption that the rooting depth of phreatophytes is constant, due to a lack of temporal rooting depth data. This assumption contrasts with studies that have shown the ability of plants to adapt to changes in groundwater levels (Fan et al., 2017; Robinson, 1958).

Although we've noted a decline in groundwater contribution to Groundwater Dependent Ecosystems (GDEs), we haven't explicitly factored in potential impacts from climate change or unsustainable groundwater extraction on GDE extent. Also, we can't conclude on GDE habitat loss solely from our findings, as we haven't observed a consistent lack of groundwater contribution throughout the year. The potential underestimation of groundwater level changes (Figure S5) and withdrawals at a high resolution (Figure S6) in our simulations could be a contributing factor.

In future work we intend to apply our framework to the global scale and better assess the individual impacts of groundwater withdrawals and climate change on the extent of GDEs under different scenarios. This would also require us to translate the change in degree of groundwater contribution to a change in GDE extent. This work will be accompanied by improved hydrogeological schematization and better calibration methods, with the aim to provide a good basis for ecological assessments, where changes in GDE extent are linked to changes in species richness.

In summary, the framework introduced in this study represents a GDE mapping approach that allows the assessment of spatio-temporal dynamics associated with the dependency of ecosystems on groundwater resources. This generic methodological framework not only enhances our understanding of the spatial distribution of GDEs but also establishes a foundation for interdisciplinary research between ecology and hydrology. By offering a global perspective on hotspot areas of GDEs under various hydroclimatic conditions, this methodology can inform decision-making processes regarding groundwater allocation and species conservation efforts. Such initiatives are crucial for advancing the objectives outlined in for example the Kunming-Montréal Global Biodiversity Framework and Sustainable Development Goal 15, which aims to halt biodiversity loss.



**Author contribution**

NGO, MFPB and ES designed the study. NGO performed the analyses, validation and visualisation of the results under the supervision of ES, MTHvV, AS and MFPB. NGO developed the methodological framework in close collaboration with ES. NGO wrote the original manuscript draft and all co-authors reviewed and edited the manuscript.

**Code/Data availability**

Code for running GLOBGM can be found at https://github.com/UU-Hydro/GLOBGM. Model outputs are available upon request.

**Competing interests**

The contact author has declared that none of the authors has any competing interests.

**Financial support**

NGO and MFPB acknowledge support from the ERC Advanced Grant scheme (Grant no. 101019185 – GEOWAT)





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
