# Peer review of "Mapping groundwater dependent ecosystems using a high-resolution global groundwater model"

_Hydrology and Earth System Sciences, 2024_

## Referee Comment (RC3)

Comments on hess-2024-112

**1. general comments**

This paper investigates a method for mapping the potential continental-scale distribution of GDEs using a global groundwater model, which aligns with the scope of HESS. The framework and steps for mapping GDEs are clearly delineated. Although there are some discrepancies between the results and the actual atlas, the findings are still acceptable for large-scale research, and the results are adequate to support the interpretations and conclusions. Unfortunately, the data and parameters necessary for coupling large-scale hydrological models with groundwater models—such as precipitation, evaporation, groundwater extraction, infiltration zoning, aquifer characteristics, and groundwater levels—are not yet clearly presented. Additionally, the impact of climate change and human activities on groundwater recharge has not been thoroughly analyzed. There has been insufficient discussion comparing the application of modeling methods with other remote sensing techniques. This indicates that the paper requires further revision and enhancement.

**2. specific comments**

**(1) Abstract**

The manuscript states that at the end of the Abstract section "The proposed framework and methodology provide a basis for analyzing how global impacts of climate change and water use may affect GDEs extent and health", However, there appears to be no data or analysis of outcomes regarding climate change and water use to support this claim.

**(2) Introduction**

Since remote sensing is used as one of the effective methods in both regional and continental scale for identifying the potential GDEs distribution, why does the author abandon the approach and instead directly employ the surface water-groundwater model method for simulation and mapping of GDEs? Why not combine remote sensing method with surface-groundwater modeling method together?

**(3) Data and methodology**

This section outlines three steps of the GDEs mapping framework with illustrations and figures. However, the third step in Fig.1 presents the content of static GDEs mapping and analysis, which appears to lack the transient GDEs mapping analysis.

**Section 2.1 Defining GDE classes (step 1):** The author defines that the saturated area fraction greater than 50% and a shallow groundwater table (less than 5m) classified as a wetland GDEs. What is the scientific basis for that?

**Section 2.2 Model set-up, sensitivity analysis and output evaluation (step 2):** This section does not explore how the conceptual models of hydrology and groundwater are constructed, particularly how the boundary conditions of continental-scale groundwater models are established, how the permeability coefficients of phreatic aquifers, confined aquifers and riverbeds are acquired, and how the boundaries and hydraulic connections between the adjacent basins or hydrogeological units are determined. Is it necessary to mesh-refinement so that the conductance data from riverbeds be utilized in groundwater models? In determining net recharge, how can data be obtained on evaporation for aquatic area, wetland and terrestrial area, as well as groundwater infiltration due to precipitation? For the Pcr-globwb-2 model to simulate the saturated area fraction, which soil parameters or parameters from the unsaturated zone and saturated zone need to be input?

**Section 2.3 GDE mapping (step 3):**

When selecting transient GDE mapping, why are the two time periods 1979-1999 and 2000-2020 chosen? Are they related to climate change (such as changes in precipitation) and shifts in human activity (such as groundwater extraction)?

**(4) Results**

**Groundwater depth** is a crucial parameter for determining the GDEs, particularly the terrestrial GDEs, for example depth of 5m just mentioned in the paper, as it most directly reflects the distribution of GDEs. Why not select the typical years from the period of 1979-1999 and 2000-2020 to create a contour map of groundwater depth and compare it with the atlas that has already been produced?

**This paper examines** the contribution of groundwater to the stream in the Murray-Darling basin. However, how can you explain the decline in the dependency

ratio when both groundwater levels and stream flow are decreasing? Were the monitoring sites for groundwater levels and stream flow selected from the upper, middle, or lower reaches of the basin? We are unsure whether the simulation accuracy at the watershed-scale will be higher than that at the continental-scale. Why not to map the distribution of GDEs at the basin-scale for typical years and compare the results with the actual atlas and then get a hit rate?

**(5) Discussions and conclusions**

The method discussed in this paper still exhibits a notable gap in evaluation accuracy when compared to the actual GDEs distribution derived from the Australian atlas. Can it be utilized for assessments at other regional, continental, and even global scales? What are the advantages and disadvantages of this method in relation to other scholars' combined remote sensing hydrogeological survey techniques? Has it been compared and analyzed against the relevant results of the following article? ——Rohde, M.M., Albano, C.M., Huggins, X. *et al.* Groundwater-dependent ecosystem map exposes global dryland protection needs. *Nature* **632**, 101–107 (2024). https://doi.org/10.1038/s41586-024-07702-8

3. technical corrections

Line 19: "using a hit rate, false alarm, and critical success index," Perhaps the term "missing rate" was lost. It would be changed to "using a critical success index derived from hit rate, false alarm, and missing rate"

Line 99, Fig.1: The abstract outlines a step for evaluation of transient mapping; however, Figure 1 does not provide an analysis and its arrow indication is unclear. The name of fig.1 is "Groundwater dependent ecosystems (GDE) modelling framework or mapping framework?

Line 110-111: Please confirm it is that the maximum rooting depth is greater than the depth to groundwater table.

Line 229: groundwater level variation ∽80% with a relative variance < 0. 6). _ missing a bracket.

Figure 3: Lack of scale bar

Line 261: false alarm ratio, or false alarm rate?

**Line 262-263:** and green represents hit rate.

Line 285 Figure 5: The figure is unclear and lacks a scale bar.

Line 294: depends on groundwater level and streamflow.

---

## Author Response (AR1)

**Reviewer 1 Gleeson**

Overall, I think this is an interesting, important and worthwhile manuscript. I appreciate the purpose (improved temporal-resolution modeling of different GDE types) as well as the method (coupled hydrologic model) and geographic focus (Australia where there is good data), and the results seem reasonable. I have a few critiques of the methods that I think would improve the manuscript.

We thank Reviewer 1 for his kind remarks about our paper.

I hate suggesting to include a few contributions that I have been a part of but i can't see anyway around this. This is a recent overview of groundwater and ecosystems that could provide more background on terminology and processes: Gleeson, T., Huggins, X., Vázquez Suñé, E., Arrojo-Agudo, P., Connor, R. (2022) Groundwater and Ecosystems. Chapter 6 of Groundwater: Making the invisible visible, UNESCO World Water Development Report.

Answer: Thanks for this useful reference. We have duly referenced it in the manuscript.

It would be good to at least mention that you are not covering subsurface ecosystems. We took a similar approach to mapping terrestrial and aquatic ecosystems in this: Huggins, X., Gleeson, T., Serrano, D., Zipper, S., Jehn, F., Rohde, M.M., Abell, R., Vigerstol, K., Hartmann, A. (2023) Overlooked risks and opportunities in groundwatersheds of the world's protected areas. Nature Sustainability.

Answer We mention subsurface GDEs in our introduction and will add a justification for not including them in our revised manuscript. We have also included the suggested reference.

We used an inference based terrestrial and aquatic inference-based approach to map terrestrial GDEs, lentic aquatic GDEs and lotic aquatic GDEs. Based on these, my most significant critique is dividing GDEs in lentic (non-flowing; lakes/wetlands) vs. lotic (flowing; rivers, streams) rather than aquatic vs wetland. Ecologists often differentiate this way since they function very differently.

Answer: We acknowledge that this would be a useful further subdivision of aquatic ecosystems, especially from an ecological perspective. There are of course several ways of dividing ecosystems into different classes. The division that we chose puts the groundwater-dependent part first, following Foster et al. (2006) and also fits the Australian GDE atlas (Doody et al. 2017). In this sense, both lakes and streams belong to aquatic ecosystems. Wetlands ecosystems are mapped based on permanent soil saturation ('Wetlands') and shallow groundwater depth (groundwater dependent), whereas 'Terrestrial groundwater ecosystems' are mapped based on root interactions with groundwater depth.

**References**

Foster S, Koundouri P, Tuinhof A, Kemper K, Nanni M and Garduño N 2006 *Groundwater* Dependent Ecosystems the Challenge of Balanced Assessment and Adequate Conservation Briefing Note Series 15 (Washington DC: GW·MATE/Worldbank)

Doody, T. M., Barron, O. V., Dowsley, K., Emelyanova, I., Fawcett, J., Overton, I. C., ... & Warren, G. (2017). Continental mapping of groundwater dependent ecosystems: A methodological framework to integrate diverse data and expert opinion. *Journal of Hydrology: Regional Studies*, *10*, 61-81.

This critique leads to my next concern: defining aquatic ecosystem dependency as the ratio to groundwater discharge to streamflow. This might be appropriate for rivers or streams but

is inappropriate for lakes which may have very little flow but still may groundwater dependent.

Answer: In our modelling approach, lakes do have flows and, due to their position (mostly in convergence zones and alluvial plains) and size, are subject to significant groundwater inflow. Hence, our classification approach also works for lakes. Nevertheless, we have excluded them from our analyses for two main reasons. First, for determining groundwater dependence, lake volume is likely more important than flow and absolute lake volumes are not explicitly included in our model. In addition, we cannot account for other complexities such as groundwater influence on lake stratification. Also in the revised manuscript, we will make more clear that we focus on lotic aquatic systems and exclude lenthic aquatic systems this case being lakes.

Two more methodological concerns:

5 m is deep water table as minimum for wetlands seems too deep and needs more justification to me..

Answer: We believe that a groundwater table threshold < 5 meters effectively captures groundwater dependent wetlands. While groundwater levels closer to the surface (0.5 to 3 meters) support core wetland functions (Eamus et al., 2006; Winter, 1999), wetlands in arid and semi-arid regions can still exhibit groundwater dependence with water tables up to 5 meters, particularly in peripheral or drought-adapted areas (Stromberg et al., 2010). The 5 m threshold thus accommodates both core and peripheral groundwater-supported zones across varied climates and is consistent with topographic variations and uncertainties in surface elevations within a 10x10 km grid cell. The 50% soil saturation was added to discern dry areas with shallow groundwater levels from actual wetlands that are typically situated in areas with topographic convergence. We also performed sensitivity analysis varying groundwater depth and saturated area fraction with a total of 100 combinations, which shows this threshold produces the highest critical success index when validating against the Australian GDE atlas .We will add this description of the rationale behind our threshold to the supplementary of the revised manuscript.

- Eamus, D., Froend, R., Loomes, R., Hose, G., & Murray, B. (2006). A functional methodology for determining the groundwater regime needed to maintain the health of groundwaterdependent vegetation. *Australian Journal of Botany*, *54*(2), 97-114.
- Stromberg, J., Lite, S., & Dixon, M. (2010). Effects of stream flow patterns on riparian vegetation of a semiarid river: implications for a changing climate. *River research and applications*, *26*(6), 712-729.

Winter, T. C. (1999). Relation of streams, lakes, and wetlands to groundwater flow systems. *Hydrogeology Journal*, *7*, 28-45.

I was also surprised that the difference between known and likely GDEs was not distinguished. I would test the importance of this assumption.

Answer: Thank you for the suggestion. Attached is the outcome of comparing our mapped GDEs to known GDEs as reported by the Australian Atlas.

Table 1 : Summary statistics of similarity matrix of mapped GDEs against known GDEs in the Austrialian GDE atlas by Doody et al. 2017

| GDE         | Hit rate (Known and Likely
GDEs) | Hit rate (Known GDEs) |
|-------------|-------------------------------------|-----------------------|
| Aquatic     | 87 %                                | 79 %                  |
| Wetland     | 95%                                 | 90 %                  |
| Terrestrial | 92 %                                | 89 %                  |

**Referee 2**

This scientific work is very interesting and well-structured. It focuses on the study of groundwater-dependent ecosystems (GDEs) in Australia under both static and transient conditions, with a methodological approach based on coupling two models (PCR-GLOBWB 2 and MODFLOW). The objective is to identify these GDEs, comparing them with the official atlases of this nation, and also study their temporal evolution at a fine resolution. It uses a large dataset of observed data to calibrate the simulated levels.

Asnwer: We thank the reviewer for the kind words.

My only remaining concern is the high error between the observed and simulated level data. In Lines 225-226 the authors stated: "Our simulated heads are deeper than the observed, however with ~ 70 % having a bias ranging from 0 to 5 m...The relative variance shows an underestimation of groundwater level variation ~ 80% with a relative variance < 0. 6." In my opinion, the error is not small if it can be as large as a 5 m difference. For a coastal aquifer it is a large and significant error that can delineate a different flow system. In the transient condition where you want to get an idea of how the whole system evolves over time, such a large error can delineate totally different scenarios. One could better argue this result in the discussion section, although it has already been mentioned as a limitation.

Answer: This is indeed a valid point. However, we are confident that the results of the groundwater model are useful for GDE-mapping, given that we found a much lower bias (we refer to the SI, Figure S3) and better evaluation statistics for shallower groundwater levels. To clarify, we will also separately report the validation statistics for shallow groundwater levels (< 10 m) to show that these are accurate enough for mapping.

Another question is about how the authors calibrate and validate the first model (PCR-GLOBWB 2), because the results of this model represent the input data in MODFLOW, therefore the overall results rely on the accuracy of the output data in the first stage.

Answer: Thanks for bringing this up. In fact, we did not perform a calibration exercise in this paper, and still, we obtain acceptable results. The reason that we do not calibrate the model to our study area is that we want to test our approach for its global accuracy, including for data scarce regions. We will focus more on calibrating the models in a follow-up study with a global extent, using groundwater level data that are now becoming more globally available.

In the following, a few suggestions:

- Line 41: insert the two citations into the same parenthesis
- Lines 104-105: avoid repetition of the word "only"
- Lines 130-131: is the second model used MODFLOW 2005? If yes, please specify it in this sentence
- Lines 205-206: repetition of saturated word
- Line 236: delete the double point at the end of the sentence
- Numbering of figures should agree with their appearance in the text. This also applies to supplementary material. For example, in the results, figure S3 is commented on first before S1 or S2
- It could be better if the authors add in the maps some toponyms mentioned in the text like Tasmania, Great Artesian basin, Murray Darlin Basin, New South Wales for a better understanding of the readers
- Line 171: "If there were multiple wells within a 1km cell, we take the average of these." Considering the same month/year?

Answer: Thanks for these suggestions. We will implement these suggested changes in a revised version of the paper.

**Referee 3**

**1.general comments**

This paper investigates a method for mapping the potential continental-scale distribution of GDEs using a global groundwater model, which aligns with the scope of HESS. The framework and steps for mapping GDEs are clearly delineated. Although there are some discrepancies between the results and the actual atlas, the findings are still acceptable for large-scale research, and the results are adequate to support the interpretations and conclusions.

Asnwer: We thank the reviewer for these encouraging remarks.

Unfortunately, the data and parameters necessary for coupling large-scale hydrological models with groundwater models—such as precipitation, evaporation, groundwater extraction, infiltration zoning, aquifer characteristics, and groundwater levels—are not yet clearly presented.

Answer: We did not present this information because it is available in several of our previous publications that we refer to (Sutanudjaja et al. 2018; De Graaf et al. (2017), Verkaik et al. (2023). However, we agree that it would be informative for the reader to provide more context. Therefore, we will:

- 1. Add a table to the SI with an overview of the datasets and their source. These will include meteorological forcing, land use, soil, soil physical parameters, the surface water network, groundwater extraction estimates, aquifer characteristics, and groundwater level data.
- 2. To illustrate spatial variability in forcing and parameter fields of the groundwater model, we will add to the SI: maps of average rainfall, average potential evapotranspiration, average groundwater recharge, groundwater extraction for the last year, and conductivity and thickness of confining layer and the aquifer. We will add a reference to the groundwater model data repository (Verkaik et al., 2023) where these time series of the forcing (monthly time step) and the parameter fields can be obtained.

**Additionally, the impact of climate change and human activities on groundwater recharge has not been thoroughly analyzed.**

Answer: We concur that climate change and human activities have impact on the GDE occurrence that we observe. Therefore, we: a) analyse the impacts of groundwater table trends and wet/dry periods on aquatic ecosystems; b) we quantify the differences in groundwater recharge between two periods (1979-1999 and 1999-2019) and compare these to the identified patterns of changes in GDEs (S5 and 6). We include additional information to make this clearer in the revised paper. We did not analyse the impacts of climate change and human impacts separately using a full factorial analysis, because we consider this out of the scope of this paper. The scope of this paper is to test a mapping method and its sensitivity to changes in groundwater depth over time. We believe that this has been sufficiently covered by our analyses. In subsequent work we will focus more on how past and future changes in GDE extent relate to climate signals and changes in human water use.

**There has been insufficient discussion comparing the application of modeling methods with other remote sensing techniques.**

Answer: We stress that other techniques are mentioned in the introduction section, among which remote sensing. We focus on modelling, since we aim to project future changes in GDE extent in a follow-up study, which is not possible with remote sensing.

**2.specific comments**

**(1) Abstract**

The manuscript states that at the end of the Abstract section "The proposed framework and methodology provide a basis for analyzing how global impacts of climate change and water use may affect GDEs extent and health", However, there appears to be no data or analysis of outcomes regarding climate change and water use to support this claim.

Answer: We note that we do provide a tentative assessment of impacts of both climate change (reflected in groundwater recharge change) and groundwater pumping, as explained in our reply to one of the main comments above. Nevertheless, we will formulate the claim more carefully as:

"The proposed framework and methodology provide a first step towards a method for analysing how impacts of climate change and human water use may affect the extent and condition of GDEs globally."

**(2) Introduction**

Since remote sensing is used as one of the effective methods in both regional and continental scale for identifying the potential GDEs distribution, why does the author abandon the approach and instead directly employ the surface water-groundwater model method for simulation and mapping of GDEs? Why not combine remote sensing method with surface-groundwater modeling method together?

Answer: The main reason for using a modelling approach, as stated in the Introduction, is that we intended to develop a mapping method that can be used in a next step (follow up work) to project GDE change in the future. This cannot be done using remote sensing alone. We could however use remote sensing data for evaluation and calibration, and we do this in a sense, since the Australian GDE Atlas that we evaluate our approach on is partly based on remote sensing (in addition to field surveying and expert knowledge) (Doody et al. 2017).

**(3) Data and methodology**

This section outlines three steps of the GDEs mapping framework with illustrations and figures. However, the third step in Fig.1 presents the content of static GDEs mapping and analysis, which appears to lack the transient GDEs mapping analysis.

Answer: For transient, we split our simulation time into two periods of 20 years each (1-1-1979 to 31-12-1999 and 1-1-2000 to 01-01-2019) and we analysed the changes in the average number of months that GDEs depend on groundwater between period 2 (1-1-2000 to 01-01-2019) and period 1 (1-1-1979 to 31-12-1999). Here we kindly refer to Figures 5, 6 and 7.

**Section 2.1 Defining GDE classes (step 1)** : The author defines that the saturated area fraction greater than 50% and a shallow groundwater table (less than 5m) classified as a wetland GDEs. What is the scientific basis for that ?

Thanks for this comment. As explained also in our reply to one of the comments of Referee 1, a groundwater table threshold < 5 meters effectively captures groundwater dependent wetlands as levels closer to the surface (0.5 to 3 meters) support core wetland functions (Eamus et al., 2006; Winter, 1999). In arid and semi-arid regions, wetlands can still exhibit groundwater dependence with water tables up to 5 meters, particularly in peripheral or drought-adapted areas (Stromberg et al., 2010). This threshold accommodates both core and peripheral groundwater-supported zones across varied climates and is consistent with topographic variations and uncertainties in surface elevations within a 10x10 km grid cell. While the 50% soil saturation was added to discern dry areas with shallow groundwater

levels from actual wetlands that are typically situated in areas with topographic convergence. We also performed sensitivity analysis varying groundwater depth and saturated area fraction with a total of 100 combinations which shows this threshold produces the highest critical success index when validating against the Australian GDE atlas.

Eamus, D., Froend, R., Loomes, R., Hose, G., & Murray, B. (2006). A functional methodology for determining the groundwater regime needed to maintain the health of groundwater-dependent vegetation. *Australian Journal of Botany*, *54*(2), 97-114. Stromberg, J., Lite, S., & Dixon, M. (2010). Effects of stream flow patterns on riparian vegetation of a semiarid river: implications for a changing climate. *River research and applications*, *26*(6), 712-729.

Winter, T. C. (1999). Relation of streams, lakes, and wetlands to groundwater flow systems. *Hydrogeology Journal*, *7*, 28-45.

**Section 2.2 Model set-up, sensitivity analysis and output evaluation (step 2)** : This section does not explore how the conceptual models of hydrology and groundwater are constructed, particularly how the boundary conditions of continental-scale groundwater models are established, how the permeability coefficients of phreatic aquifers, confined aquifers and riverbeds are acquired, and how the boundaries and hydraulic connections between the adjacent basins or hydrogeological units are determined. Is it necessary to mesh-refinement so that the conductance data from riverbeds be utilized in groundwater models? In determining net recharge, how can data be obtained on evaporation for aquatic area, wetland and terrestrial area, as well as groundwater infiltration due to precipitation? For the Pcr-globwb-2 model to simulate the saturated area fraction, which soil parameters or parameters from the unsaturated zone and saturated zone need to be input?

Here we kindly refer to our answer to the general question above about the input data and parameterization.

**Section 2.3 GDE mapping (step 3) :**

When selecting transient GDE mapping, why are the two time periods 1979-1999 and 2000-2020 chosen? Are they related to climate change (such as changes in precipitation) and shifts in human activity (such as groundwater extraction)?

Answer: We have 40 years of simulation. We split this period in half to estimate if changes could be picked up by the mapping method and if these could be explained from groundwater level changes subject to recharge and pumping. The change in recharge between the two periods is not necessarily related to climate change, but the average climate could be different between the periods due to the occurrence of large climate modus (e.g. ENSO). Furthermore, pumping is larger in the second half due to socioeconomic development.

**(4) Results**

**Groundwater depth** is a crucial parameter for determining the GDEs, particularly the terrestrial GDEs, for example depth of 5m just mentioned in the paper, as it most directly reflects the distribution of GDEs. Why not select the typical years from the period of 1979-1999 and 2000-2020 to create a contour map of groundwater depth and compare it with the atlas that has already been produced?

Answer: We will add a more extensive justifation of the threshold of 5 m (see also our replies to earlier comments on this point). We show the average groundwater depth map and the

differences in groundwater depth between the first and the second period in the SI, as well as comparison with observations.

**This paper examines** the contribution of groundwater to the stream in the Murray-Darling basin. However, how can you explain the decline in the dependency ratio when both groundwater levels and stream flow are decreasing?

Answer: If groundwater discharge is decreasing more rapidly than streamflow, the ratio declines. What happens to the ratio is thus very much dependent on how drought or desiccation impacts the groundwater system and surface runoff. This can be different from one place to the next and is captured by our modelling system.

Were the monitoring sites for groundwater levels and stream flow selected from the upper, middle, or lower reaches of the basin?

Answer: We use all the monitoring sites with sufficient data for the evaluation for the groundwater levels simulations. In the example we use examples from the lower reaches in the MD basin. We will show the location of the monitoring stations on a map in the revised paper.

We are unsure whether the simulation accuracy at the watershed-scale will be higher than that at the continental-scale. Why not to map the distribution of GDEs at the basin-scale for typical years and compare the results with the actual atlas and then get a hit rate?

Answer: We are not sure whether we understand this remark. The Murray-Darlin results are extracted from the continental scale simulation and are thus not necessarily more accurate.

**(5) Discussions and conclusions**

The method discussed in this paper still exhibits a notable gap in evaluation accuracy when compared to the actual GDEs distribution derived from the Australian atlas. Can it be utilized for assessments at other regional, continental, and even global scales? What are the advantages and disadvantages of this method in relation to other scholars' combined remote sensing hydrogeological survey techniques?

Has it been compared and analyzed against the relevant results of the following article? — Rohde, M.M., Albano, C.M., Huggins, X. *et al.* Groundwater-dependent ecosystem map exposes global dryland protection needs. *Nature* **632**, 101–107 (2024). https://doi.org/10.1038/s41586-024-07702-8

Answer: We respectfully beg to accept that we disagree about the accuracy of our method, which we consider quite high when looking at the high hit rates, low miss rates and high critical success ratios. We really appreciate the paper by Rohde et al. (2024), which we also noted ourselves. This paper was not published at the time we submitted our paper and we will now refer to it in the revised version. When comparing their approach to ours, we note the following:

- o Their evaluation results in Australia are similar to our evaluation results
- This is again a method based on stationary mapping and not suitable to project future changes of GDE-occurrence;

We will add a short discussion on comparison of our results with their findings.

**3.technical corrections**

**Line 19:** "using a hit rate, false alarm, and critical success index," Perhaps the term "**missing rate**" was lost. It would be changed to "using a critical success index derived from hit rate, false alarm, and missing rate"

Line 99, Fig.1: The abstract outlines a step for evaluation of transient mapping; however, Figure 1 does not provide an analysis and its arrow indication is unclear. The name of fig.1 is "Groundwater dependent ecosystems (GDE) modelling framework or mapping framework? Line 110-111: Please confirm it is that the maximum rooting depth is greater than the depth to groundwater table.

Line 229: groundwater level variation  $\sim$ 80% with a relative variance < 0. 6). \_ missing a bracket.

Figure 3: Lack of scale bar

Line 261: false alarm ratio, or false alarm rate?

Line 262-263: and green represents hit rate.

Line 285 Figure 5: The figure is unclear and lacks a scale bar.

Line 294: depends on groundwater level and streamflow.

Answer: Thanks for these suggestions. We will implement these suggested changes in a revised version of the paper.

---

## Author Response (AR2)

**Authors response**

We thank the editor for the suggested alterations to our rebuttal. We are sorry that we did not make it clearer where and what our suggested changes were. We hope that this updated complete rebuttal to all the reviewers' comments suits better to track which changes we made.

**Reviewer 1**

**Overall, I think this is an interesting, important and worthwhile manuscript. I appreciate the purpose (improved temporal-resolution modelling of different GDE types) as well as the method (coupled hydrologic model) and geographic focus (Australia where there is good data), and the results seem reasonable. I have a few critiques of the methods that I think would improve the manuscript.**

 *Response: We thank Reviewer 1 for his kind remarks about our paper.*

**I hate suggesting to include a few contributions that I have been a part of but i can't see anyway around this. This is a recent overview of groundwater and ecosystems that could provide more background on terminology and processes: Gleeson, T., Huggins, X., Vázquez Suñé, E.. Arrojo-Agudo, P., Connor, R. (2022) Groundwater and Ecosystems. Chapter 6 of Groundwater: Making the invisible visible, UNESCO World Water Development Report.**

*Response: Thanks for this useful reference. We have duly referenced it in the introduction section of the manuscript in the introduction*

*lines 45-50 "The degree of dependency of GDEs on groundwater varies with ecosystem type, geology, season, aquifer type, flow paths and catchment land use (Tomlinson & Boulton, 2010; Gleeson et al., 2023)"*

**It would be good to at least mention that you are not covering subsurface ecosystems.**

**We took a similar approach to mapping terrestrial and aquatic ecosystems in this: Huggins, X., Gleeson, T., Serrano, D., Zipper, S., Jehn, F., Rohde, M.M., Abell, R., Vigerstol, K., Hartmann, A. (2023) Overlooked risks and opportunities in groundwatersheds of the world's protected areas. Nature Sustainability.**

**We used an inference based terrestrial and aquatic inference-based approach to map terrestrial GDEs, lentic aquatic GDEs and lotic aquatic GDEs. Based on these, my most significant critique is dividing GDEs in lentic (non-flowing; lakes/wetlands) vs. lotic (flowing; rivers, streams) rather than aquatic vs wetland. Ecologists often differentiate this way since they function very differently**.

*Response : We mention subsurface GDEs in our introduction and will add a justification for not including them in our revised manuscript section 2.1 (Defining GDE classes) Lines 110- 115 "Also, note that we focus on inland ecosystems only. Finally, we do not consider subsurface ecosystems that rely on groundwater, such as stygofauna communities (Foster et al., 2010) because of the complexity of mapping these communities similarly as previously done by Huggins et al., (2023) as well.". We have also included the suggested reference in section 2.1 as well (As shown above).*

*We acknowledge that this would be a useful further subdivision of aquatic ecosystems, especially from an ecological perspective. There are several ways of dividing ecosystems into different classes. The division that we chose puts the groundwater-dependent part first, following Foster et al. (2006) and also fits the Australian GDE atlas (Doody et al. 2017). In this sense, both lakes and streams belong to aquatic ecosystems. Wetlands ecosystems are mapped based on permanent soil saturation ('Wetlands') and shallow groundwater depth (groundwater dependent), whereas 'Terrestrial*

groundwater ecosystems' are mapped based on root interactions with groundwater depth. See reference of this choice of classification in the introduction

*Lines 45-50 specifically "The degree of dependency of GDEs on groundwater varies with ecosystem type, geology, season, aquifer type, flow paths and catchment land use (Tomlinson & Boulton, 2010 ; Gleeson et al., 2023). In arid and semi-arid regions, groundwater is usually a major source of water for most ecosystems. GDE types include surface water systems (aquatic GDEs, which include rivers and lakes) that rely on groundwater discharge (Kløve et al., 2011), and groundwater dependent wetlands and terrestrial ecosystems (e.g. vegetation like phreatophytes) that tap into groundwater as a source of water (Robinson, 1958)."*

*References*

*Foster S, Koundouri P, Tuinhof A, Kemper K, Nanni M and Garduño N 2006 Groundwater Dependent Ecosystems the Challenge of Balanced Assessment and Adequate Conservation Briefing Note Series 15 (Washington DC: GW·MATE/Worldbank)*

*Doody, T. M., Barron, O. V., Dowsley, K., Emelyanova, I., Fawcett, J., Overton, I. C., ... & Warren, G. (2017). Continental mapping of groundwater dependent ecosystems: A methodological framework to integrate diverse data and expert opinion. Journal of Hydrology: Regional Studies, 10, 61-81.*

**This critique leads to my next concern: defining aquatic ecosystem dependency as the ratio to groundwater discharge to streamflow. This might be appropriate for rivers or streams but is inappropriate for lakes which may have very little flow but still may groundwater dependent.**

*Response: In our modelling approach, lakes do have flows and, due to their position (mostly in convergence zones and alluvial plains) and size, are subject to significant groundwater inflow. Hence, our classification approach also works for lakes. Nevertheless, we have excluded them from our analyses for two main reasons. First, for determining groundwater dependence, lake volume is likely more important than flow and absolute lake volumes are not explicitly included in our model. In addition, we cannot account for other complexities such as groundwater influence on lake stratification. Also in the revised manuscript, we will make clearer that we focus on lotic aquatic systems and exclude lentic aquatic systems this case being lakes specifically section 2.1 (Defining GDE classes)*

*Lines 110 – 115 "We categorize groundwater dependent ecosystems into three classes based on interaction with groundwater (see Figure 2). These include: 1) ecosystems that depend on sufficient groundwater discharge (aquatic GDEs such as streams and rivers); 2) ecosystems that need shallow groundwater tables and soil saturation (wetland GDEs); 3) ecosystems that depend on groundwater for root water uptake (terrestrial GDEs with phreatophyte vegetation). In case of aquatic ecosystems, we do not include lakes due to the complexities in determining the contribution of groundwater in lentic systems. "*

[Figure]

*Figure 1: Criteria for defining groundwater dependent ecosystems, with $Q_{gw}$ = local groundwater discharge, $Q$ = accumulated stream flow, $G_{wt}$ = groundwater table depth, $F_{sat}$ = saturated area fraction and $dR_{max}$ = maximum rooting depth.*

**Two more methodological concerns:**

**5 m is deep water table as minimum for wetlands seems too deep and needs more justification to me.**

*Response: We believe that a groundwater table threshold < 5 meters effectively captures groundwater dependent wetlands. While groundwater levels closer to the surface (0.5 to 3 meters) support core wetland functions (Eamus et al., 2006; Winter, 1999), wetlands in arid and semi-arid regions can still exhibit groundwater dependence with water tables up to 5 meters, particularly in peripheral or drought-adapted areas (Stromberg et al., 2010). The 5 m threshold thus accommodates both core and peripheral groundwater-supported zones across varied climates and is consistent with topographic variations and uncertainties in surface elevations within a 10x10 km grid cell., The 50% soil saturation was added to discern dry areas with shallow groundwater levels from actual wetlands that are typically situated in areas with topographic convergence. We also performed sensitivity analysis varying groundwater depth and saturated area fraction with a total of 100 combinations, which shows this threshold produces the highest critical success index when validating against the Australian GDE atlas. This justification has been included under section 2.1*

*Lines 135 – 140 "We define wetland GDEs based on the fraction of saturated area (soil wetness) and groundwater level. Any cell that has a saturated area fraction ($F_{sat}$) greater than 50% and a groundwater table depth less than 5m is classified as a wetland GDE. While groundwater levels closer to the surface (0.5 to 3 meters) support core wetland functions (Eamus et al., 2006; Winter, 1999), wetlands in arid and semi-arid regions can still exhibit groundwater dependence with water table depths up to 5 meters, particularly in peripheral or drought-adapted areas (Stromberg et al., 2010). Hence, the threshold of 5m accommodates both core and peripheral groundwater-supported zones across varied climates. We have added the 50% soil saturation threshold to discern dry areas with shallow groundwater levels from actual wetlands, which are typically saturated at the surface. We performed a sensitivity analysis for varying groundwater depth thresholds (1 to 5 m) and saturated area fractions (0.1 to 1.0) with a total of 100 combinations, showing that the threshold of 5 meters produces the highest critical success index when validating against the Australian GDE atlas*

*(See Supplementary Figure S1). In the latter case, we assessed the "degree of groundwater dependency" for each GDE type identified on a monthly time step (Figure 1)."*

[Figure]

*Figure S1. Sensitivity analysis of threshold selection for wetlands against hit-rate, false alarm rate and critical success index with the Australian GDE atlas*

**I was also surprised that the difference between known and likely GDEs was not distinguished. I would test the importance of this assumption**.

*Response: Thank you for the suggestion. Attached is the outcome of comparing our mapped GDEs to known GDEs as reported by the Australian Atlas. This table has also been included in the supplementary of the manuscript* *page 4* *(Table S4).*

*Table S4: Summary statistics of similarity matrix of mapped GDEs against known GDEs in the Australian GDE atlas by Doody et al. 2017*

| GDE | Hit rate (Known and Likely GDEs) | Hit rate (Known GDEs) |
|---|---|---|
| Aquatic | 87 % | 79 % |
| Wetland | 95% | 90 % |
| Terrestrial | 92 % | 89 % |

Eamus, D., Froend, R., Loomes, R., Hose, G., & Murray, B. (2006). A functional methodology for determining the groundwater regime needed to maintain the health of groundwater-dependent vegetation. *Australian Journal of Botany*, 54(2), 97-114.

Stromberg, J., Lite, S., & Dixon, M. (2010). Effects of stream flow patterns on riparian vegetation of a semiarid river: implications for a changing climate. *River research and applications*, 26(6), 712-729.

Winter, T. C. (1999). Relation of streams, lakes, and wetlands to groundwater flow systems. *Hydrogeology Journal*, 7, 28-45.

**Reviewer 2**

**This scientific work is very interesting and well-structured. It focuses on the study of groundwater-dependent ecosystems (GDEs) in Australia under both static and transient conditions, with a methodological approach based on coupling two models (PCR-GLOBWB 2 and MODFLOW). The objective is to identify these GDEs, comparing them with the official atlases of this nation, and also study their temporal evolution at a fine resolution. It uses a large dataset of observed data to calibrate the simulated levels.**

*Response: We thank the reviewer for the kind words.*

**My only remaining concern is the high error between the observed and simulated level data. In Lines 225-226 the authors stated: "Our simulated heads are deeper than the observed, however with ~ 70 % having a bias ranging from 0 to 5 m. The relative variance shows an underestimation of groundwater level variation ~ 80% with a relative variance < 0. 6." In my opinion, the error is not small if it can be as large as a 5 m difference. For a coastal aquifer it is a large and significant error that can delineate a different flow system. In the transient condition where you want to get an idea of how the whole system evolves over time, such a large error can delineate totally different scenarios. One could better argue this result in the discussion section, although it has already been mentioned as a limitation.**

*Response: This is indeed a valid point. However, we are confident that the results of the groundwater model are useful for GDE-mapping, given that we found a much lower bias and better evaluation statistics for shallower groundwater levels. To clarify, we will also separately report the validation statistics for shallow groundwater levels (< 10 m)* *(figure S6)* *in the supplementary to show that these are accurate enough for mapping.*

[Figure]

*Figure S6. Histogram of mean bias of categorized observed groundwater depths (m).*

**Another question is about how the authors calibrate and validate the first model (PCR-GLOBWB 2), because the results of this model represent the input data in MODFLOW, therefore the overall results rely on the accuracy of the output data in the first stage.**

*Response: Thanks for bringing this up. In fact, we did not perform a calibration exercise in this paper, and still, we obtain acceptable results. The reason that we do not calibrate the model to our study area is that we want to test our approach for its global accuracy, including for data scarce regions. We will focus more on calibrating the models in a follow-up study with a global extent, using groundwater level data that are now becoming more globally available.*

**In the following, a few suggestions:**

- **Line 41: insert the two citations into the same parenthesis**
- **Lines 104-105: avoid repetition of the word "only"**
- **Lines 130-131: is the second model used MODFLOW 2005? If yes, please specify it in this sentence**
- **Lines 205-206: repetition of saturated word**
- **Line 236: delete the double point at the end of the sentence**
- ***Numbering of figures should agree with their appearance in the text. This also applies to supplementary material. For example, in the results, figure S3 is commented on first before S1 or S2***
- **It could be better if the authors add in the maps some toponyms mentioned in the text like Tasmania, Great Artesian basin, Murray Darlin Basin, New South Wales for a better understanding of the readers**
- **Line 171: "If there were multiple wells within a 1km cell, we take the average of these." Considering the same month/year?**

*Response: Thanks for these suggestions. We will implement these suggested changes in a revised version of the paper. The technical and typological mistakes have been rectified. The second model used is GLOBGM and been specified in Lines 150- 155 "The second is a time-dependent (transient) groundwater flow model, GLOBGM (Verkaik et al., 2022). ". The figures have been re-numbered chronologically. For the evaluation of the model we have reconstructed the sentence to "If there were multiple wells within a 1km cell, we calculate the average of these considering the same year." in lines 195 – 200.*

**Reviewer 3**

**1. General comments**

**This paper investigates a method for mapping the potential continental-scale distribution of GDEs using a global groundwater model, which aligns with the scope of HESS. The framework and steps for mapping GDEs are clearly delineated. Although there are some discrepancies between the results and the actual atlas, the findings are still acceptable for large-scale research, and the results are adequate to support the interpretations and conclusions.**

*Response: We thank the reviewer for these encouraging remarks.*

**Unfortunately, the data and parameters necessary for coupling large-scale hydrological models with groundwater models—such as precipitation, evaporation, groundwater extraction, infiltration zoning, aquifer characteristics, and groundwater levels—are not yet clearly presented.**

*Response: We did not present this information because it is available in several of our previous publications that we refer to (Sutanudjaja et al. 2018; De Graaf et al. (2017), Verkaik et al. (2023). However, we agree that it would be informative for the reader to provide more context. Therefore, we have:*

1. *Add a table to the SI (Table S1) with an overview of the datasets and their source. These will include meteorological forcing, land use, soil, soil physical parameters, the surface water network, groundwater extraction estimates, aquifer characteristics, and groundwater level data.*
2. *To illustrate spatial variability in forcing and parameter fields of the groundwater model, we will add to the SI (Figures S3, S9 and S10): maps of average rainfall, average potential evapotranspiration, average groundwater recharge, groundwater extraction for the last year, and conductivity and thickness of confining layer and the aquifer. We will add a reference to the groundwater model data repository (Verkaik et al., 2023) where these time series of the forcing (monthly time step) and the parameter fields can be obtained.*

*Table S1: Model input and parameters as well as data sources*

| Description | Symbol | Unit | References/Sources |
|---|---|---|---|
| *Upper and lower soil store parameters* | | | *FAO (2007) soil map; van Beek and Bierkens (2009)* |
| • *Soil thickness* | *Z1 and Z2* | *m* | |
| • *Residual soil moisture content* | *θr1 and θr2* | *m³/m³* | |
| • *Soil moisture at saturation* | *θs1 and θs2* | *m³/m³* | |
| • *Soil water storage capacity per soil layer: SC = Z/(θs − θr)* | *SC1 and SC2* | *m* | |
| • *Soil matric suctions at saturation* | *ψs1 and ψs2* | *m* | |
| • *Exponent in the soil water retention curve* | *β1 and β2* | *dimensionless* | |
| • *Saturated hydraulic conductivities of upper and lower soil stores* | *K1 and K2* | *m/day* | |
| • *Total soil water storage capacities = SCupp + SClow* | *Wmax* | *m* | |
| *Land cover fraction: land cover areas (including extent of irrigated areas) over cell areas* | *flcov* | *m²/m²* | *GLCC v2.0 map (USGS, 1997); Olson (1994a, b); MIRCA2000 dataset (Portmann et al., 2010); FAOSTAT (2012)* |
| *Topographical parameters* | | | *HydroSHEDS (Lehner et al., 2008); Hydro1k (Verdin and Greenlee, 1996); GTOPO30 (Gesch et al., 1999)* |

| | Symbol | Units | Reference |
|---|---|---|---|
| • Cell-average DEM | $DEM_{avg}$ | m | |
| • Floodplain elevation | $DEM_{fpl}$ | m | |
| Root fractions per soil layer | $Rf_{upp}$ and $Rfl_{ow}$ | dimensionless | Canadell et al. (1996); van Beek and Bierkens (2009) |
| Arno scheme (Todini, 1999; Hagemann and Gates, 2003): exponents defining soil water capacity distribution | $\beta_{arno}$ | dimensionless | Canadell et al. (1996); Hagemann et al. (1999); Hagemann et al. (2002); van Beek (2008); van Beek and Bierkens (2009) |
| Ratio of cell-minimum soil storage to Wmax | $fw_{min}$ | m/m | van Beek (2008); van Beek and Bierkens (2009) |
| Ratio of cell-maximum soil storage to Wmax | $fw_{max}$ | m/m | van Beek (2008); van Beek and Bierkens (2009) |
| Parameters related to phenology | | | Hagemann et al. (1999); Hagemann (2002); van Beek (2008); van Beek and Bierkens (2009) |
| • Crop coefficient | $K_c$ | dimensionless | |
| • Interception capacity | $S_{int-max}$ | m | |
| • Vegetation cover fraction | $C_v$ | m²/m² | |
| Groundwater parameters | | | GLHYMPS map (Gleeson et al., 2014); van Beek (2008); van Beek and Bierkens (2009) |
| Aquifer transmissivity | KD | m²/day | |
| Aquifer specific yield | Sy | m³/m³ | |
| Groundwater recession coefficient | $J^{-1}$ | day⁻¹ | |
| Meteorological forcing | | | van Beek (2008); CRU (Harris et al., 2014); W5E5 (Cucchi et al, 2020; Stefan et al, 2021) |
| • Total precipitation | P | m/day | |
| • Atmospheric air temperature | $T_{air}$ | °C or K | |
| • Reference potential evaporation and transpiration | $E_{ref,pot}$ | m/day | |
| Others | | | |
| • Non-irrigation sectoral water demand (i.e., livestock, domestic, and industrial) | - | m/day | Wada et al. (2014) |
| • Desalinated water | - | m/day | Wada et al. (2011a); FAO (2016) |
| • Lakes and reservoirs | - | - | GLWD1 (Lehner and Döll, 2004); GRanD (Lehner et al., 2011) |

[Figure]

*Figure S3: Confining layer thickness, average recharge, average annual total precipitation, and average annual total potential evapotranspiration Hydraulic conductivity, aquifer thickness.*

[Figure]

*Figure S9. Differences below surface level (m) in average simulated (top) and observed (bottom) groundwater depths between 1979-2000 and 2001-2019. Red areas indicate a decline in groundwater levels and blue areas indicate areas with an increase in groundwater levels. White areas in bottom figure are locations with no well data.*

[Figure]

Difference in recharge (m/year)

[Figure]

Abstraction (m/year)

*Figure S10: Changes between simulated groundwater recharge and between period 2001-2019 relative 1979-2000 and average abstraction rates over 2001-2019.*

**Additionally, the impact of climate change and human activities on groundwater recharge has not been thoroughly analyzed.**

*Response: We concur that climate change and human activities have impact on the GDE occurrence that we observe. Therefore, we: a) analyse the impacts of groundwater table trends and wet/dry periods on aquatic ecosystems; b) we quantify the differences in groundwater recharge between two periods (1979-1999 and 1999-2019)*

*Lines 300 – 310 "For aquatic GDEs, we assessed temporal changes in the different dependency ratio $\frac{Q_{gw}}{Q}$ categories. We observe that there is a decline in the average number of months in all dependency classes ecosystems (Figure 5) and that the decline in groundwater contribution is mostly observed in streams in the Murray Darlin Basin. This is in accordance with the decline in groundwater levels between the two periods in both the simulations and the observations (Figure S9). It is important to realize that the dependency ratio depends on both the groundwater depth and related groundwater discharge $Q_{gw}$and the streamflow itself. This is illustrated in Figure 6 that shows simulated time series of $\frac{Q_{gw}}{Q}$, groundwater depth and total streamflow. The figure shows that the groundwater levels are constrained at the top by the drainage system and also shows the intermittent character of the Australian climate, with wet periods alternating with dry periods where groundwater levels decline, and streamflow becomes almost zero. The top figure shows a negative trend in groundwater levels. However, streamflow is also declining, offsetting the decline in groundwater discharge, resulting in a smaller negative trend in groundwater dependency $\frac{Q_{gw}}{Q}$. The zoom at the bottom shows the importance of discharge variability. November 2005 and July 2006 show almost the same shallow water table. However, streamflow peaks in November 2005, which makes for a low dependency ratio, while the 2006 streamflow is low in July, making the dependency on groundwater discharge large"*

*and compare these to the identified patterns of changes in GDEs (Figures S5 and 6). We include additional information to make this clearer in the revised paper lines 405- 410 "Although we noted a decline in groundwater contribution to Australian GDEs over the past decades, we have not explicitly factored in potential impacts from climate change or unsustainable groundwater extraction on GDE extent.".*

*We did not analyse the impacts of climate change and human impacts **separately** using a full factorial analysis, because we consider this out of the scope of this paper. The scope of this paper is to test a mapping method and its sensitivity to changes in groundwater depth over time. We did however compare patterns of change in simulated groundwater level changes with patterns of drivers. See the text on lines 315-325: "We have added some additional analyses to improve understanding of the drivers' groundwater level changes between both periods. Figure S10 shows the difference in simulated groundwater recharge between the periods 2001-2019 relative to 1979-2000 and the simulated groundwater withdrawal over the 2001-2019 period. The changes in groundwater recharge reflect the impact of climate variability and/or change on the groundwater system while the locations with groundwater withdrawal reflect the direct human impacts. A thorough factor analysis is beyond the scope of this study, but a comparison of Figures 7 with S9 suggests that climate variability mainly explains the changes in groundwater depth in North, Central and Western Australia while both factors play a role in Eastern Australia." We believe that this has been sufficiently covered by our analyses. In subsequent work we will focus more on how past and future changes in GDE extent relate to climate signals and changes in human water use.*

**There has been insufficient discussion comparing the application of modeling methods with other remote sensing techniques.**

*Response: We stress that other techniques are mentioned in the introduction section, among which remote sensing from* lines 60 to 70 *"Until the past decade, mapping of GDEs was predominantly done at local scales, through laborious and costly methods that involved long hours of field surveys (Eamus et al., 2006; Hatton & Evans, 1998). More recently, GDEs have also been mapped based on satellite imagery such as MODIS (Castellazzi et al., 2019). Some large-scale satellite imagery-based mapping studies (> 50km) have been done in Chile (Duran-Llacer et al., 2022), Colorado and Nevada (Werstak et al., 2012), California (Howard & Merrifield, 2010), The Netherlands (Bonte et al., 2013; Hoogland et al., 2010), Ireland (Kilroy et al., 2009), South Africa (Münch & Conrad, 2007), Spain (Martínez-Santos et al., 2021; Münch & Conrad, 2007) and Australia (Barron et al., 2014; Brim Box et al., 2022; Glanville et al., 2016). The first continental mapping was done for Australia (Doody et al., 2017), combining remote sensing, GIS and expert knowledge to create a GDE atlas for the continent.*

*"* We focus on modelling, since we aim to project future changes in GDE extent in a follow-up study which is not possible with remote sensing. (lines 415-420) *"In future work we intend to apply our framework to the global scale and better assess the individual impacts of groundwater withdrawals and climate change on the extent of GDEs under different scenarios. This would also require us to translate the change in degree of groundwater contribution to a change in GDE extent. This work will be accompanied by improved hydrogeological schematization and better calibration methods, with the aim to provide a good basis for ecological assessments, where changes in GDE extent are linked to changes in species richness."*

**2. Specific comments**

**(1)Abstract**

**The manuscript states that at the end of the Abstract section "The proposed framework and methodology provide a basis for analyzing how global impacts of climate change and water use may affect GDEs extent and health", However, there appears to be no data or analysis of outcomes regarding climate change and water use to support this claim.**

*Response: We note that we do provide a tentative assessment of impacts of both climate change (reflected in groundwater recharge change) and groundwater pumping, as explained in our reply to one of the main comments above. Nevertheless, we will formulate the claim more carefully as:*

*"The proposed framework and methodology provide a first step towards a method for analysing how impacts of climate change and human water use may affect the extent and condition of GDEs globally."* lines 20-25.

**(2) Introduction**

**Since remote sensing is used as one of the effective methods in both regional and continental scale for identifying the potential GDEs distribution, why does the author abandon the approach and instead directly employ the surface water-groundwater model method for simulation and mapping of GDEs? Why not combine remote sensing method with surface-groundwater modeling method together?**

*Response: The main reason for using a modelling approach, as stated in the Introduction, is that we intended to develop a mapping method that can be used in a next step (follow up work) to project GDE change in the future. This cannot be done using remote sensing alone. We could however use remote sensing data for evaluation and calibration, and we do this in a sense, since the Australian GDE Atlas that we evaluate our approach (Figure 3 in the manuscript) on is partly based on remote sensing (in addition to field surveying and expert knowledge) (Doody et al. 2017).*

[Figure]

Figure 2: Mapped GDEs based on steady state groundwater model results evaluated against the Australian GDE atlas showing hit rate, false alarm rate, miss rate and the CSI for the three GDE classes. Blue colour represents missed ecosystems, dark red represents false alarm and green represents hit rate.

**(3) Data and methodology**

**This section outlines three steps of the GDEs mapping framework with illustrations and figures. However, the third step in Fig.1 presents the content of static GDEs mapping and analysis, which appears to lack the transient GDEs mapping analysis.**

*Response: For transient, we split our simulation time into two periods of 20 years each (1-1-1979 to 31-12-1999 and 1-1-2000 to 01-01-2019) and we analysed the changes in the average number of months that GDEs depend on groundwater between period 2 (1-1-2000 to 01-01-2019) and period 1 (1-1-1979 to 31-12-1999). Here we kindly refer to Figures 5, 6 and 7.*

[Figure]

*Figure 3: Change in groundwater dependency of aquatic GDEs between 1979-2000 and 2001-2019; (a) Maps of the direction of change in the average number of months that aquatic GDEs depend on groundwater; the left figure shows the change in the number of months $\frac{Q_{gw}}{Q} > 0$ (low to high dependency), the middle figure $\frac{Q_{gw}}{Q} > 0.5$ (moderate to high dependency) and the right figure $\frac{Q_{gw}}{Q} > 0.75$ (high dependency); Red areas indicate a decrease in the average number of months with groundwater dependency and blue indicates an increase; (b) associated frequency distributions of change in number of months.*

[Figure]

[Figure]

*Figure 6: Example time series of $\frac{Q_{gw}}{Q}$ for a downstream river reach location in the Darling River (location indicated in the aquatic GDE map on top). Top: time series of simulated $\frac{Q_{gw}}{Q}$ , total streamflow (Q) and groundwater depth, including trendlines. Right: zoom into a selected timeframe (green bar in the left figure) to show how the variability of dependence of $\frac{Q_{gw}}{Q}$ depends on groundwater level and streamflow.*

[Figure]

*Figure 7: Change in average number of months of groundwater the dependency of terrestrial GDEs (phreatophytes) and wetland GDEs; a) direction of change in terrestrial GDEs (phreatophytes); b) direction of change wetland GDEs. Red areas indicate a decrease in the average number of months with groundwater dependency, green indicates no change between the periods and blue indicates an increase; (c) and (d) associated cumulative frequency distributions of change in number of months.*

**Section 2.1 Defining GDE classes (step 1):The author defines that the saturated area fraction greater than 50% and a shallow groundwater table (less than 5m) classified as a wetland GDEs. What is the scientific basis for that?**

*Response: As stated earlier, we believe that a groundwater table threshold < 5 meters effectively captures groundwater dependent wetlands. While groundwater levels closer to the surface (0.5 to 3 meters) support core wetland functions (Eamus et al., 2006; Winter, 1999), wetlands in arid and semi-arid regions can still exhibit groundwater dependence with water tables up to 5 meters, particularly in peripheral or drought-adapted areas (Stromberg et al., 2010). The 5 m threshold thus accommodates both core and peripheral groundwater-supported zones across varied climates and is consistent with topographic variations and uncertainties in surface elevations within a 10x10 km grid cell. This justification has been included under section 2. 1* *lines 135 – 140 "We define wetland GDEs based on the fraction of saturated area (soil wetness) and groundwater level. Any cell that has a saturated area fraction ($F_{sat}$) greater than 50% and a groundwater table depth less than 5m is classified as a wetland GDE. While groundwater levels closer to the surface (0.5 to 3 meters) support core wetland functions (Eamus et al., 2006; Winter, 1999), wetlands in arid and semi-arid regions can still exhibit groundwater dependence with water table depths up to 5 meters, particularly in peripheral or drought-adapted areas (Stromberg et al., 2010). Hence, the threshold of 5m accommodates both core and peripheral groundwater-supported zones across varied climates. We*

*added the 50% soil saturation threshold to discern dry areas with shallow groundwater levels from actual wetlands, which are typically saturated at the surface. We performed a sensitivity analysis for varying groundwater depth thresholds (1 to 5 m) and saturated area fractions (0.1 to 1.0) with a total of 100 combinations, showing that the threshold of 5 meters produces the highest critical success index when validating against the Australian GDE atlas (See Supplementary Figure S1). In the latter case, we assessed the "degree of groundwater dependency" for each GDE type identified on the basis of a monthly time step (Figure 1)."*

[Figure]

*Figure S1. Sensitivity analysis of threshold selection for wetlands against hit-rate, false alarm rate and critical success index with the Australian GDE atlas*

*The 50% soil saturation was added to discern dry areas with shallow groundwater levels from actual wetlands that are typically situated in areas with topographic convergence. We also performed sensitivity analysis varying groundwater depth and saturated area fraction with a total of 100 combinations, which shows this threshold produces the highest critical success index when validating against the Australian GDE atlas .We will add this description of the rationale behind our threshold to the supplementary of the revised manuscript (Figure S1)*

*Eamus, D., Froend, R., Loomes, R., Hose, G., & Murray, B. (2006). A functional methodology for determining the groundwater regime needed to maintain the health of groundwater-dependent vegetation. Australian Journal of Botany, 54(2), 97-114.*

*Stromberg, J., Lite, S., & Dixon, M. (2010). Effects of stream flow patterns on riparian vegetation of a semiarid river: implications for a changing climate. River research and applications, 26(6), 712-729.*

*Winter, T. C. (1999). Relation of streams, lakes, and wetlands to groundwater flow systems. Hydrogeology Journal, 7, 28-45.*

**Section 2.2 Model set-up, sensitivity analysis and output evaluation (step 2):This section does not explore how the conceptual models of hydrology and groundwater are constructed, particularly how the boundary conditions of continental-scale groundwater models are established, how the permeability coefficients of phreatic aquifers, confined aquifers and riverbeds are acquired, and how the boundaries and hydraulic connections between the adjacent basins or hydrogeological units are determined. Is it necessary to mesh-refinement so that the conductance data from riverbeds be utilized in groundwater models? In determining net recharge, how can data be obtained on evaporation for aquatic area, wetland and terrestrial area, as well as groundwater infiltration due to precipitation? For the Pcr-globwb-2 model to simulate the saturated area fraction, which soil parameters or parameters from the unsaturated zone and saturated zone need to be input?**

*Response: Here we kindly refer to our answer to the general question above about the input data and parameterization.*

**(4) Results**

**Groundwater depth is a crucial parameter for determining the GDEs, particularly the terrestrial GDEs, for example depth of 5m just mentioned in the paper, as it most directly reflects the distribution of GDEs. Why not select the typical years from the period of 1979-1999 and 2000-2020 to create a contour map of groundwater depth and compare it with the atlas that has already been produced?**

*Response: We have added a more extensive justification of the threshold of 5 m (see also our replies to earlier comments on this point). We show the average groundwater depth map and the differences in groundwater depth between the first and the second period in the supplementary figure S9, as well as comparison with observations. We will add this description of the rationale behind our threshold to the supplementary of the revised manuscript (Figure S1) also as seen above.*

**This paper examines the contribution of groundwater to the stream in the Murray-Darling basin. However, how can you explain the decline in the dependency ratio when both groundwater levels and stream flow are decreasing?**

*Response: If groundwater discharge is decreasing more rapidly than streamflow, the ratio declines. What happens to the ratio is thus very much dependent on how drought or desiccation impacts the groundwater system and surface runoff. This can be different from one place to the next and is captured by our modelling system. Lines 300 to 310 "We observe that there is a decline in the average number of months in all dependency classes ecosystems (Figure 5) and that the decline in groundwater contribution is mostly observed in streams in the Murray Darlin Basin. This is in accordance with the decline in groundwater levels between the two periods in both the simulations and the observations (Figure S9). It is important to realize that the dependency ratio depends on both the groundwater depth and related groundwater discharge $Q_{gw}$ and the streamflow itself. This is illustrated in Figure 6 that shows simulated time series of $\frac{Q_{gw}}{Q}$, groundwater depth and total streamflow. The figure shows that the groundwater levels are constrained at the top by the drainage system and also shows the intermittent character of the Australian climate, with wet periods alternating with dry periods where groundwater levels decline, and streamflow becomes almost zero".*

**Were the monitoring sites for groundwater levels and stream flow selected from the upper, middle, or lower reaches of the basin?**

*Response: We use all the monitoring sites with sufficient data for the evaluation for the groundwater levels simulations. In the example we use examples from the lower reaches in the MD basin. We will show the location of the monitoring stations on a map in the revised paper (Figure 6) in the manuscript (also found above in response to other comments).*

*We are unsure whether the simulation accuracy at the watershed-scale will be higher than that at the continental-scale. Why not to map the distribution of GDEs at the basin-scale for typical years and compare the results with the actual atlas and then get a hit rate?*

*We are not sure whether we understand this remark. The Murray-Darlin results are extracted from the continental scale simulation and are thus not necessarily more accurate.*

(5)**Discussions and conclusions**

**The method discussed in this paper still exhibits a notable gap in evaluation accuracy when compared to the actual GDEs distribution derived from the Australian atlas. Can it be utilized for assessments at other regional, continental, and even global scales? What are the advantages and disadvantages of this method in relation to other scholars' combined remote sensing hydrogeological survey techniques?**

**Has it been compared and analyzed against the relevant results of the following article? ——Rohde, M.M., Albano, C.M., Huggins, X. *et al.* Groundwater-dependent ecosystem map exposes global dryland protection needs. *Nature* 632, 101–107 (2024). https://doi.org/10.1038/s41586-024-07702-8**

*Response: We respectfully beg to accept that we disagree about the accuracy of our method, which we consider quite high when looking at the high hit rates, low miss rates and high critical success ratios. We really appreciate the paper by Rohde et al. (2024), which we also noted ourselves. This paper was*

*not published at the time we submitted our paper and we will now refer to it in the revised version. When comparing their approach to ours, we note the following:*

- o *Their evaluation results in Australia are similar to our evaluation results* *(Figure 4 )* *in the manuscript*
- o *This is again a method based on stationary mapping and not suitable to project future changes of GDE-occurrence.*

**3.technical corrections**

**Line 19:** "using a hit rate, false alarm, and critical success index," Perhaps the term **"missing rate"** was lost. It would be changed to "using a critical success index derived from hit rate, false alarm, and missing rate"

**Line 99, Fig.1:** The abstract outlines a step for evaluation of transient mapping; however, Figure 1 does not provide an analysis and its arrow indication is unclear. The name of fig.1 is "Groundwater dependent ecosystems (GDE) **modelling** framework or **mapping** framework?

**Line 110-111:** Please confirm it is that the maximum rooting depth is greater than the depth to groundwater table.

Line 229: groundwater level variation ⌣80% with a relative variance < 0. 6). _ **missing a bracket**.

**Figure 3:** Lack of scale bar

**Line 261:** false alarm ratio, or false alarm rate?

**Line 262-263:** and green **represents** hit rate.

Line 285 Figure 5: The figure is unclear and lacks a scale bar.

Line 294: depends on groundwater level **and** streamflow.

*Response: Thanks for these suggestions.*

*In* *line 110-111* *now* *120 to 125* *"the maximum rooting depth should be less than the groundwater table". We have implemented all these suggested changes in a revised version of the paper. We have rectified all he technical changes. Figure 3 and Figure 5 have a legend.*